# Mechanisms of thrombin-Induced myometrial contractions: Potential targets of progesterone

**Fumitomo Nishimura, Haruta Mogami* , Kaori Moriuchi, Yoshitsugu Chigusa, Masaki Mandai, Eiji Kondoh**

Department of Gynecology and Obstetrics, Kyoto University, Graduate School of Medicine, Kyoto, Japan

* mogami@kuhp.kyoto-u.ac.jp

**Data Availability Statement:** All relevant data are within the manuscript and its Supporting Information files, and Supplementary Video 1 is

## Abstract

Intrauterine bleeding during pregnancy is a major risk factor for preterm birth. Thrombin, the most abundant coagulation factor in blood, is associated with uterine myometrial contraction. Here, we investigated the molecular mechanism and signaling of thrombin-induced myometrial contraction. First, histologic studies of placental abruption, as a representative intrauterine bleeding, revealed that thrombin was expressed within the infiltrating hemorrhage and that thrombin receptor (protease-activated receptor 1, PAR1) was highly expressed in myometrial cells surrounding the hemorrhage. Treatment of human myometrial cells with thrombin resulted in augmented contraction via PAR1. Thrombin-induced signaling to myosin was then mediated by activation of myosin light chain kinase- and Rho-induced phosphorylation of myosin light chain-2. In addition, thrombin increased *prostaglandin-endoperoxidase synthase-2* (*PTGS2* or *COX2*) mRNA and prostaglandin E2 and F2α synthesis in human myometrial cells. Thrombin significantly increased the mRNA level of interleukine-1β, whereas it decreased the expressions of prostaglandin EP3 and F2α receptors. Progesterone partially blocked thrombin-induced myometrial contractions, which was accompanied by suppression of the thrombin-induced increase of *PTGS2* and *IL1B* mRNA expressions as well as suppression of PAR1 expression. Collectively, thrombin induces myometrial contractions by two mechanisms, including direct activation of myosin and indirect increases in prostaglandin synthesis. The results suggest a therapeutic potential of progesterone for preterm labor complicated by intrauterine bleeding.

## Introduction

Intrauterine or vaginal bleeding is a risk factor for preterm birth [1]. Subchorionic hematoma in the first and second trimester doubles the preterm birth rate [2, 3], and decidual or uterine hemorrhage is a strong risk for preterm premature rupture of membranes (pPROM) [4]. In addition, placental abruption, which causes massive intrauterine bleeding in the decidual space, causes strong uterine contraction [5]. Onset of placental abruption itself is closely associated with first trimester bleeding [2, 3, 6].

uploaded to Zenodo (https://doi.org/10.5281/
zenodo.3240679).

**Funding:** The authors received no specific funding
for this work.

**Competing interests:** The authors have declared
that no competing interests exist.

Thrombin is a serine proteinase that is most abundantly contained in blood [7]. In addition to blood coagulation, thrombin plays a significant role in preterm birth [8]. Patients with preterm labor have increased plasma [9] and amniotic fluid [10] thrombin–antithrombin complex levels compared with normal pregnant women. Risk of pPROM is increased by thrombin [11]. Previously, we showed that thrombin activity was increased in human amnion tissues from women with preterm birth, and thrombin increased (i) expression and activity of matrix metalloproteinases (MMPs) and (ii) prostaglandin (PG) synthesis in primary amnion mesenchymal cells [12]. Moreover, intra-uterine injection of thrombin in pregnant mice caused preterm birth [12]. Other studies have shown that thrombin induces myometrial contractions in rats [13, 14]. The thrombin–antithrombin complex gradually rises during normal pregnancy, reaching maximum in the 3rd stage of labor [15, 16]. Therefore, dysregulation of thrombin activity has the potential to cause a premature onset of labor, leading to preterm birth.

Myosin II is the primary motor protein in muscle [17]. Myosin comprises heavy and light chains. Cellular myosin II is activated by phosphorylation of its regulatory light chain (MLC) at Ser19, which allows myosin II to interact with actin, assembling an actomyosin complex and initiation of contraction [17]. Two groups of enzymes control MLC phosphorylation. One consists of kinases that phosphorylate MLC (MLC kinase, MLCK, and Rho-associated protein kinase, ROCK), promoting activity, and the other is a phosphatase that dephosphorylates MLC, inhibiting activity [18].Throughout pregnancy, uterine quiescence is maintained by progesterone [19]. Progesterone has been used for the prevention and treatment of preterm labor, and clinical evidence of its effectiveness is accumulating [20–24]. However, the effect of progesterone on preterm labor caused by intrauterine bleeding is unclear.

In this study, we investigated the molecular mechanisms of thrombin-induced uterine smooth muscle contraction using primary human myometrial smooth muscle cells. We also tested the hypothesis that progesterone may ameliorate thrombin-induced myometrial contraction.

## Materials and methods

### Immunofluorescence of human pregnant uterus

Myometrium was obtained from two cases of placental abruption at 1) 25 weeks and 5 days and 2) 33 weeks and 4 days with written informed consent. Hysterectomy was performed due to uncontrollable massive uterine bleeding with disseminated intravascular coagulopathy (DIC). Myometrium was fixed in 10% formaldehyde, and then paraffin embedded. Antigen retrieval was performed by incubation with proteinase K (P8107S, New England Biolab, working concentration, 0.6 units/mL) for 10 min at 37˚C. Sections were then preincubated with 10% normal goat serum (50062Z, Life Technologies) with 0.3% Triton X-100 for 30 min at room temperature. Subsequently, tissue sections were incubated with primary antibodies in PBS with 1% BSA and 0.3% Triton X-100 at 4˚C overnight. Primary antibodies used and concentration were as follows: thrombin (coagulation factor II, Novus Biologicals, NBP1-58268, Research Resource Identifier (RRID): AB_11023777, 1:100) and PAR1 (N2-11, Novus Biologicals, NBP1-71770, RRID: AB_11027203, 1:100). Thereafter, sections were incubated with Alexa Fluor 488 (Goat anti-Mouse IgG, A11001, RRID: AB_2534069, Invitrogen, 1:500 dilution) or 594-conjugated secondary antibodies (Goat anti-Rabbit IgG, A11012, Invitrogen, RRID: AB_2534079, 1:500 dilution) in 10% normal goat serum for 1 h at room temperature. Slides were mounted with Prolong Gold Antifade Reagent with DAPI (P36935, Molecular Probes). Images were taken by Leica TCX-SP8 confocal microscopy.

### Isolation and culture of human myometrial cells

Human myometrial smooth cells were isolated as previously described [25]. Briefly, ~8 g of myometrial tissue was obtained from non-pregnant premenopausal women undergoing

hysterectomy. Indications for hysterectomy were leiomyoma or endometriosis. To test the contraction of pregnant myometrial cells, myometrial tissues from the uterine fundus were obtained from a rare case of cesarean hysterectomy due to pregnancy complicated by cervical cancer stage Ib1 (S1 Fig). The tissue was minced into fragments and agitated in 60 mL of minimum essential medium eagle (MEM) containing 80 mg of collagenase B (11088807001, Roche), 40 mg of DNase I (11284932001, Roche), and 1.5 mL of 1 M HEPES for 2 h at 37˚C. Tissue was then filtered through mesh to remove non-dispersed tissue fragments. The filtrate was centrifuged at 1000 $g$ for 10 min to pellet the dispersed cells. Cells were resuspended in DMEM/F-12 that contained fetal bovine serum (10%, v/v) and antibiotic-antimycotic solution (1%, v/v). Cells were plated at a density of $0.5–1.0 \times 10^5$ cells/cm$^2$ and incubated under 20% $O_2$ and 5% CO2 at 37˚C. All experiments were repeated at least three times.

A time-lapse movie of thrombin-treated myometrial cells was as follows: myometrial cells were seeded on a plastic dish. On the 6$^{th}$ day of culture, cells were treated with 4 U/mL of thrombin or PBS. Images were acquired every minute for a period of 120 min using a live-cell imaging system (Olympus IX71N microscope with DP71 camera). Images were assembled into one frame, and a 12-s movie (×600 accelerated) using Mac Preview and iMovie (Apple) was produced and uploaded to Zenodo (https://doi.org/10.5281/zenodo.3240679).

All tissues were obtained in accordance with the Kyoto University Graduate School and Faculty of Medicine, Kyoto University Hospital Ethics Committee after obtaining written patient consent (G1149). The institute's Ethics Committee specifically approved this study.

## Collagen lattice assay

Primary human myometrial cells ($21 \times 10^5$ cells) were suspended in collagen type I mixture: 14 mL of Cellmatrix TypeI-A (0.3%, Nitta Gelatin), 1.75 mL of 10 × MEM (M0275, Sigma), and 1.75 mL of 0.08N NaOH with 200 mM HEPES. Final concentration in 6-well plates was $15 \times 10^4$ cells per well. The mixture was then placed in 6-well dishes (2 mL/well), and incubated at 37˚C for 30 min. After confirming that the gel was completely solidified, 2 mL of DMEM/F12 growth medium was overlaid. Plates were incubated for 4 to 6 days. On the day of experiments, cells in collagen gels were pretreated with the following reagents for 1 h: 100 nM PAR1 selective antagonist SCH79797 (No.1592 Tocris), 1 µM ROCK inhibitor Y-27632 (10005583, Cayman Chemical), 10 µM MLCK inhibitor, ML-7 (11801, Cayman Chemical), 10 µM of indomethacin (I-7378, Sigma), or 1 µM of progesterone (28921–64, Nacalai-tesque). After pretreatment, collagen gels were gently detached from the bottom of the well using tips of plastic pipettes, and then 2 U/mL of thrombin (T7009, Sigma) was added. Images were captured at indicated times (ChemiDoc, Biorad). When pretreatment was unnecessary, 2 U/mL of thrombin or 10 µM of PAR1 activating peptide TFLLR-NH$_2$ (1464, TOCRIS) was added after detachment of gels from the well. Each experiment was performed in triplicate, and repeated at least three times. The gel area was calculated by Image J software. Briefly, the outside of a gel was manually traced by the "Polygon selection" tool and the area was calculated by the "Measure" tool. The relative pixel area was shown in mean ± standard deviation (SD).

## Immunocytochemistry

Myometrial cells were grown in 8-well chamber slides. After thrombin treatment (2 U/mL, 30 min), cells were fixed in 4% paraformaldehyde for 10 min. Slides were incubated with 10% normal goat serum for 30 min, and treated with primary antibodies overnight at 4˚C as follows: PAR1 (N2-11, Novus Biologicals, NBP1-71770, RRID: AB_11027203, 1:100) and Phospho-Myosin Light Chain 2 (Ser19, Cell Signaling, #3671, RRID: AB_330248, 1:100). After incubation with secondary antibody (Goat anti-Rabbit IgG, Alexa Fluor 488, Invitrogen, A11008, A11008,

RRID: AB_143165, 1:500 dilution) or (Goat anti-Mouse IgG, Alexa Fluor 594, RRID: AB_2534073, Invitrogen, 1:500 dilution), slides were mounted with Prolong Gold DAPI. Images were taken by Leica TCS-SP8 confocal microscopy, and generated by Image J software.

## Immunoblots

Primary human myometrial cells were treated with 2 U/mL of thrombin for 30 min or the indicated time with or without pretreatment of SCH79797 (100 nM) or Y27632 (10 μM) for 1 h. Cells were lysed in RIPA buffer containing protease inhibitor cocktail (Complete Mini, Roche) and phosphatase inhibitor cocktail (PhosSTOP, Roche). The samples were centrifuged at 10,000 $g$ for 20 min and the supernatant were used for immunoblots. Protein concentration was assayed by Pierce BCA Protein Assay Kit (23225, Thermo Scientific) according to the manufacturer's instruction. Twenty micrograms of protein were applied to polyacrylamide gels, separated by electrophoresis, and transferred to polyvinylidene fluoride (PVDF) membranes. The membranes were blocked with 5% bovine serum albumin (BSA) for 1 h at room temperature. A membrane was incubated with primary antibodies overnight at 4°C as follows: Phospho-Myosin Light Chain 2 (Ser19, Cell Signaling, #3671, RRID: AB_330248, 1:1000), myosin light chain 2 (D18E2, Cell Signaling, #8505, RRID: AB_2728760, 1:1000), Phospho-MYPT1 (Thr696, Cell Signaling, #5163, RRID: AB_10691830, 1:1000), Phospho-MYPT1 (Thr853, Cell Signaling, #4563, RRID: AB_1031185, 1:1000) MYPT1 (D6C1, Cell Signaling, #8574, RRID: AB_10998518, 1:1000) or anti-beta actin (Abcam, ab8227, 1:2000). All of the first antibodies were diluted in 5 mL of 5% BSA/Tris Buffered Saline with Tween 20 (TBST). Thereafter, blots were incubated with secondary antibody (Goat Anti-Rabbit IgG-HRP Conjugate, #170–6515, RRID: AB_11125142, Biorad, 1:10000 in 10 mL of 5% BSA/TBST) at room temperature for 1 h. The signal was detected by chemiluminescence (Pierce ECL Plus Western Blotting Substrate, #32132, Thermo Scientific).

## Quantitative real-time Polymerase Chain Reaction (PCR)

At confluency, primary human myometrial cells were treated with the indicated dose of thrombin, IL-1β (0.1 ng/mL, 201-LB, R&D Systems), PGE2 (0.1 or 1 μM, 29334–21, Nacalaitesque), and PGF2α (P0424, Sigma) for 24 h or the indicated time. If pretreatment was necessary, cells were treated with 10 μM of indomethacin or 1 μM progesterone for 1 h.

Quantitative RT-PCR was used to determine the relative levels of gene expression [26]. Primer sequences are shown in Table 1. Gene expression was normalized to that of GAPDH which was invariant in these cells. To analyze progesterone receptor expressions, mRNA levels of total progesterone receptor (*total PgR*) and progesterone receptor isoform-B (*PgR-B*) were quantified in ng/μL of cDNA using the standard curve method (Applied Biosystems). The mRNA abundance of progesterone receptor isoform-A (*PgR-A*) was calculated by subtracting the abundance of *PgR-B* mRNA from that of *total PgR* mRNA.

## ELISA

Prostaglandin E2 and $F_{2a}$ concentration in the condition media was assayed by Parameter Prostaglandin $E_2$ Assay (KGE004B, R&D Systems) and PGF2α ELISA kit (ADI-900-069, Enzo Life Sciences) according to the manufacturer's instruction.

## Statistical analysis

Values were expressed as means ± SD. Data were analyzed by one-way analysis of variance (ANOVA) followed by the Student-Newman-Keuls test, unless otherwise indicated. *p*-values less than 0.05 were regarded as statistically significant.

**Table 1. Primer sequences used for quantitative RT-PCR.**

| GAPDH | 5'- GGAGTCAACGGATTTGGTCGTA -3' | 5'- CAACAATATCCACTTTACCAGAGTTA -3' |
|---|---|---|
| PTGS2 | 5'- GCTCAACACCGGAATTTTTGA -3' | 5'- TCGAAGGAAGGGAATGTTATTCA -3' |
| GJA1 | 5'- ACTGGCGACAGAAACAATTCTTC -3' | 5'- TTCTGCACTGTAATTAGCCCAGTT -3' |
| OXTR | 5'- GCTGCAACCCCTGGATCTAC -3' | 5'- GGAAGCGCTGCACGAGTT -3' |
| PTGER1 | 5'- CAGCCACTTCTAAGCACAACCA -3' | 5'- GAATGGCTTTTTATTCCCAAAGG -3' |
| PTGER3 | 5'- GACGGCATTCAGCTTATGG -3' | 5'- TGATGTCTGATTGAAGATCATTTTCA -3' |
| PTGFR | 5'- GAGCGGCTCCGTCTTCTG -3' | 5'- GGAGATAAAAGCCAACCACTCAA -3' |
| IL1B | 5'- TCCTGCGTGTTGAAAGATGATAA -3' | 5'- TTGGGTAATTTTTGGGATCTACACT -3' |
| PgR (total) | 5'- CGGACACCTTGCCTGAAGTT -3' | 5'- CAGGGCCGAGGGAAGAGTAG -3' |
| PgR-B | 5'- GATAAAGGAGCCGCGTGTCA -3' | 5'- GAGTACTCACAAGTCCGGCACTT -3' |

## Results

### Thrombin receptor, protease-activated receptor 1 (PAR1), is expressed in human myometrium

Expression of thrombin receptor PAR1 [7] was investigated in human myometrium. As expected, PAR1 was localized to the plasma membrane of myometrial smooth muscle cells in uterine tissue from uncomplicated pregnancy (Fig 1A) and from non-pregnant women (Fig 1B). In pregnancy, PAR1 was also strongly expressed in the decidua of human fetal membranes (Fig 1C), as previously reported [12]. Thrombin was not expressed in normal pregnant myometrium (S1 Fig). Two representative cases of placental abruption are shown in Fig 1D–1G. Placental abruption caused exudation of blood into the myometrium near the site of placental attachment (Fig 1D and 1F). Thrombin was released in this bleeding site (Fig 1E and 1G). Although not observed in the center of the hemorrhage, PAR1 was expressed in myometrial cells peripheral to the bleeding site (Fig 1E and 1G). These findings suggest that uterine contraction induced by intrauterine bleeding may involve hemorrhage-derived thrombin-myometrial PAR1 interactions.

### Thrombin increased contraction of primary human myometrial cells through PAR1

Having established PAR1 expression in myometrium and its potential relationship with uterine hemorrhage, the effect of thrombin on contraction of myometrium was quantified using collagen lattice assays and primary human myometrial cells from non-pregnant uteri.

Thrombin significantly increased the contraction of myometrial cells embedded in collagen gels from 5 min (Fig 2A). The relative pixel area with treatment of thrombin was 0.73 ± 0.02, 0.62 ± 0.02, 0.55 ± 0.03, and 0.43 ± 0.01 at 5, 15, 30, and 60 min, respectively, whereas that of PBS control was 0.90 ± 0.02, 0.82 ± 0.02, 0.78 ± 0.01, and 0.64 ± 0.04, respectively ($p$ = 0.0006, 0.0001, 0.0001, 0.0011, respectively). Similarly, thrombin induced the contraction of myometrial cells from a pregnant uterus (S2 Fig). Through time-lapse live imaging, myometrial contraction was observed in thrombin-treated cells (left panel) compared with static cells treated with PBS (right panel) (S1 Movie. https://doi.org/10.5281/zenodo.3240679).

To support a role for PAR1 in this process, PAR1 activating peptide (PAR1 AP, TFLLR) also induced time-dependent contraction of myometrial cells (Fig 2B). The relative pixel area with treatment of TFLLR was 0.74 ± 0.02, 0.58 ± 0.01, 0.57 ± 0.01, 0.57 ± 0.02, and 0.57 ± 0.02 at 5, 20, 30, 45, and 60 min, respectively, whereas that of PBS was 0.92 ± 0.03, 0.91 ± 0.04, 0.91 ± 0.03, 0.90 ± 0.02, and 0.89 ± 0.02, respectively ($p$ = 0.0001, 0.001, 0.0001, 0.0001, and 0.0001, respectively). Further, PAR1 inhibitor, SCH79797, significantly inhibited thrombin-

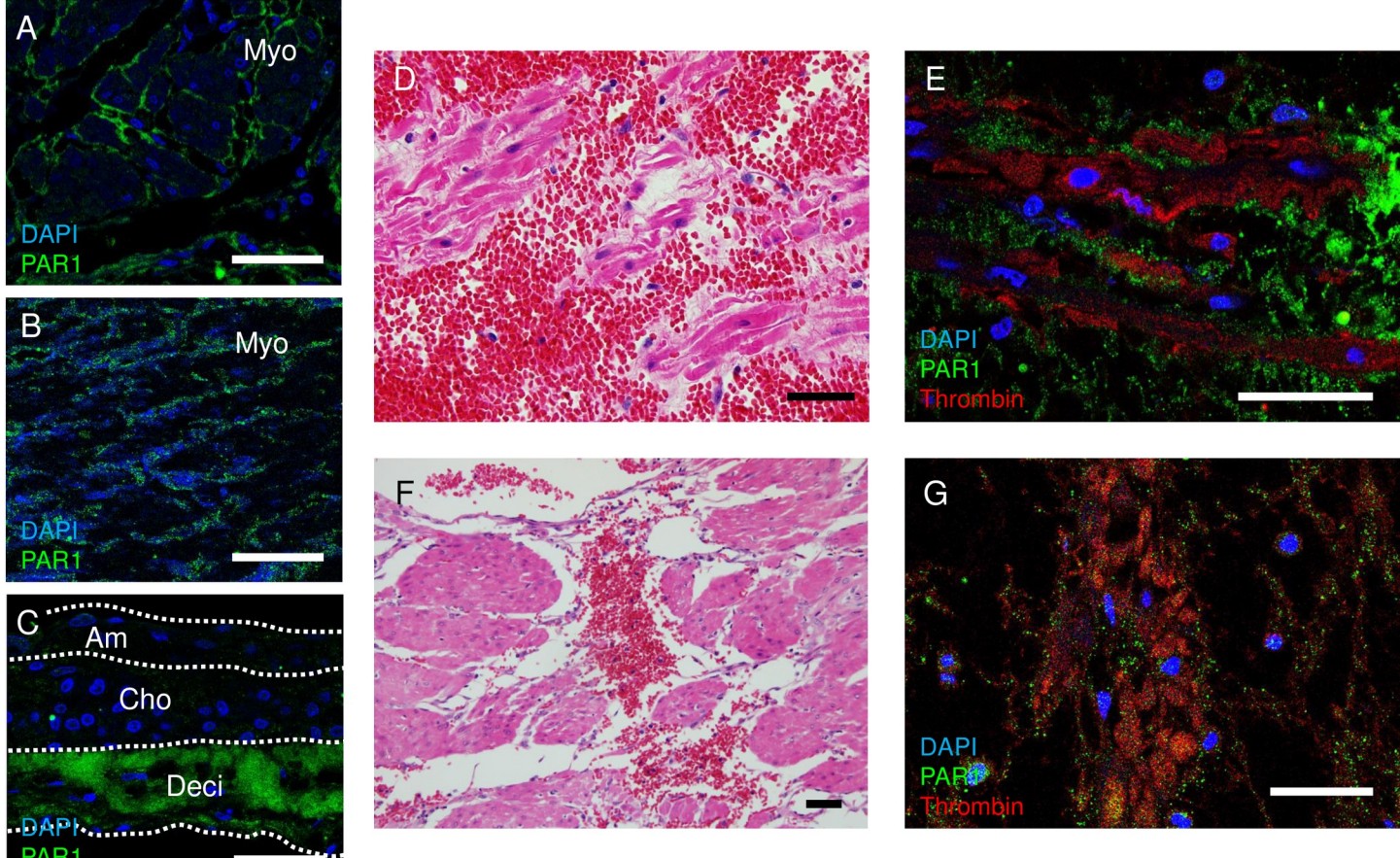

**Fig 1. Thrombin receptor, protease-activated receptor 1 (PAR1), is expressed in human myometrium. (A-C)** Immunofluorescence of PAR1 (green) in myometrium from a pregnant woman **(A)**, non-pregnant woman **(B)**, and fetal membrane **(C)**. Nuclei were stained with DAPI (blue). Am, amnion, Cho, chorion, Deci, decidua. **(D-G)** Localization of hemorrhage, thrombin, and PAR1 in placental abruption at 25 weeks of gestation **(D and E)**, and 33 weeks gestation **(F and G)** resulting in disseminated intravascular coagulopathy and uterine bleeding requiring hysterectomy for hemostasis. **(D and F)** Hematoxylin and eosin staining of the myometrium adjacent to the placenta. Note that hemorrhage infiltrated the myometrium. Bars, 50 μm. **(E and G)** Immunofluorescence of PAR1 (green), thrombin (red), and DAPI (blue) at the same location of **(A)**. Bars, 50 μm.

induced contractions of myometrial cells (Fig 2C). Thrombin treatment significantly decreased the relative pixel area compared to the control (0.68 ± 0.02 vs. 0.85 ± 0.02; $p = 0.0001$), whereas pretreatment with SCH79797 completely blocked the contraction effect of thrombin compared to thrombin alone (0.87 ± 0.01, $p = 0.0001$). SCH79797 alone inhibited the spontaneous contraction of myometrial cells-embedded gel (0.98 ± 0.01) compared to PBS control ($p = 0.0001$). These data indicate that thrombin-induced myometrial contractions are mediated via PAR1.

## Thrombin activated actin–myosin interaction by signaling phosphorylation of myosin light chain-2 (MLC2)

Activation of myosin is a key step in the actin–myosin interaction leading to myometrial contraction, and phosphorylation of serine 19 site of myosin regulatory light chain (MLC) activates myosin II protein [18]. Immunofluorescence of primary human myometrial cells revealed that PAR1 was ubiquitously expressed throughout the cell (Fig 3A). Treatment with thrombin increased phosphorylation of MLC2 significantly at 60 min (Fig 3A), and expression

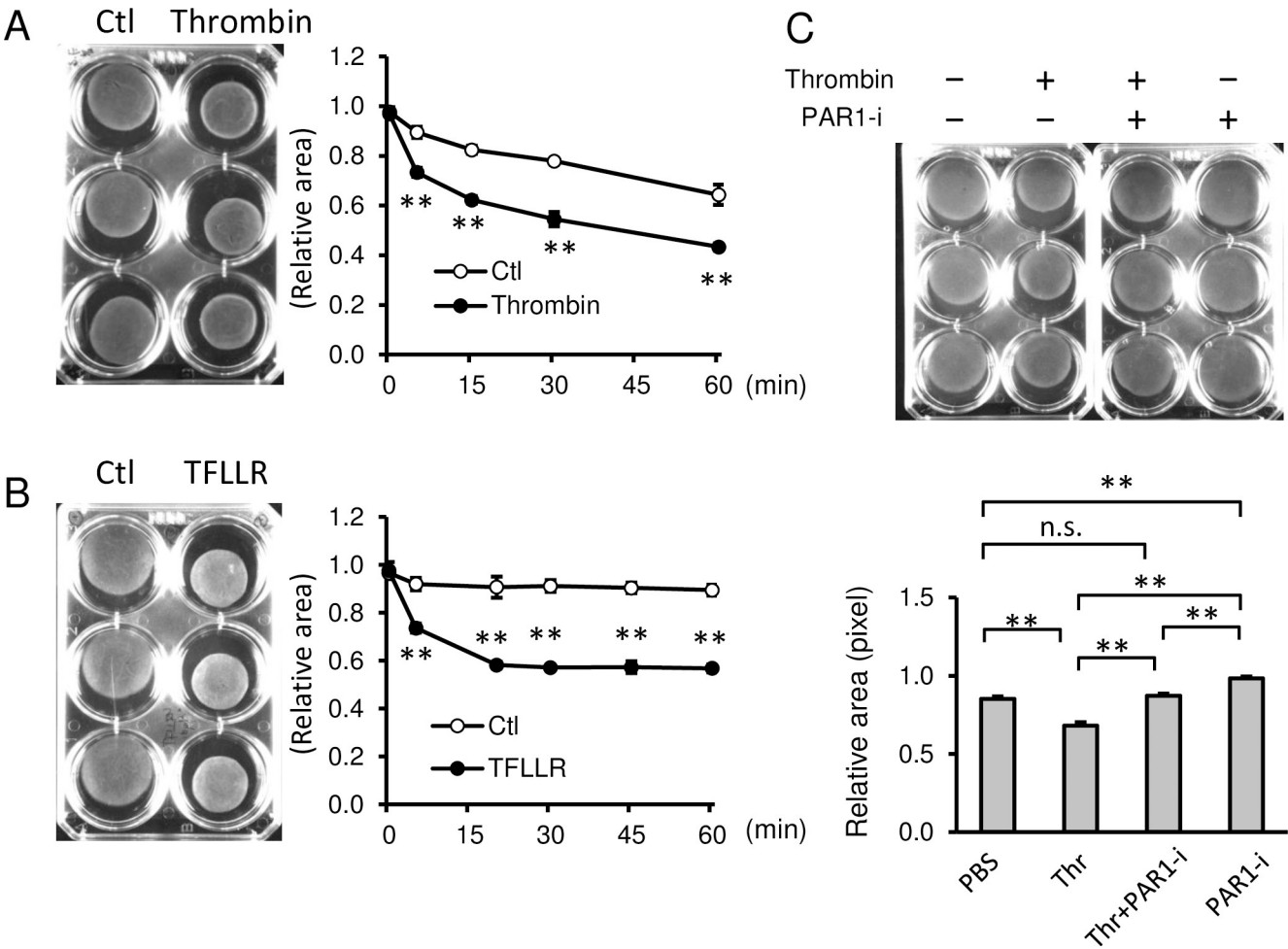

**Fig 2. Thrombin increased contraction of primary human myometrial cells through PAR1. (A and B) (Left images)** Representative images of collagen lattice assay of human myometrial cells at 30 min treated with PBS (Ctl) and thrombin **(A)** or PAR1 activating peptide, TFLLR **(B)**. **(Right graphs)** Quantification of myometrial contractions in collagen lattice assays (n = 3). **, $p < 0.01$ at each time point. **(C)** Collagen lattice assay of myometrial cells at 30 min with 2 U/mL of thrombin (Thr) pretreated with or without 100 nM PAR1 selective inhibitor (SCH79797, PAR1-i) for 1 h (n = 3). Representative image (upper panel) and quantification of gel areas (lower graph). The experiments were repeated three times. *, $p < 0.05$, and **, $p < 0.01$.

of MLC2 co-localized with PAR1 (Fig 3A merge). Immunoblot analysis revealed that thrombin increased phosphorylation of MLC2 at 5 min, which was sustained after 60 min (Fig 3B). Thereafter, dephosphorylation was initiated at 120 min (Fig 3B). PAR1 inhibitor completely blocked thrombin-induced phosphorylation of MLC2 (Fig 3C). The data suggest that thrombin binds to PAR1, resulting in activation of myosin through phosphorylation of MLC2 and contraction of myometrium.

## Thrombin activates myosin by MLCK and ROCK

Next, we investigated the mechanisms by which thrombin activates the myosin motor protein. We first tested if the myosin light chain kinase (MLCK) inhibitor ML-7 altered smooth muscle cell contraction. In collagen lattice assays, ML-7 inhibited thrombin-induced myometrial cell contraction (Fig 4A). The relative pixel area with treatment of thrombin was decreased to $0.69 \pm 0.03$ compared to that of PBS ($0.90 \pm 0.01$, $p = 0.0001$), whereas pretreatment with ML-7 blocked the contraction of thrombin ($0.86 \pm 0.03$, $p = 0.0001$). ML-7 alone inhibited the

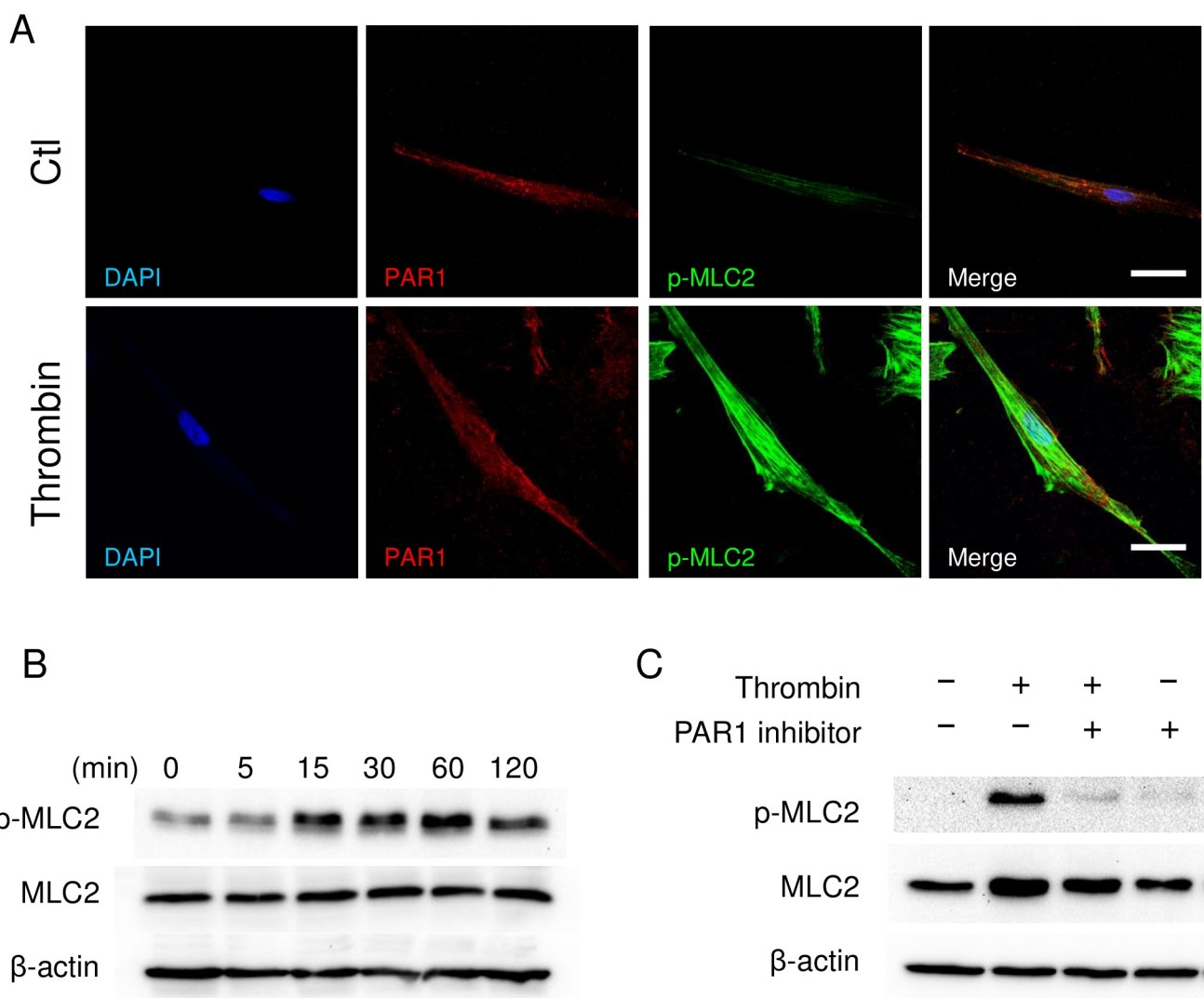

**Fig 3. Thrombin activates acto-myosin interaction by phosphorylation of myosin light chain-2 (MLC2). (A)** Immunocytochemistry of PAR1 (red) and phosphorylated MLC2 (Ser19, green) in human myometrial cells. **(B, C)** Immunoblots of phosphorylated MLC2 (p-MLC2), total MLC2, and β-actin of myometrial cells. Myometrial cells were treated with thrombin (2 U/mL) as a function of time **(B)**, or pretreated with 100 nM PAR1 inhibitor (SCH79797) for 1 h, and then treated with 2 U/mL of thrombin for 30 min **(C)**. The experiments were repeated three times.

spontaneous contraction of gel (0.97 ± 0.01) compared to PBS control ($p = 0.0487$). We next tested Rho-associated protein kinase (ROCK) inhibitor, Y-27632. Y-27632 partially inhibited the thrombin-induced contraction of myometrial cells (Fig 4B). Treatment of thrombin decreased the area of gel to 0.59 ± 0.01 compared to that of PBS (0.93 ± 0.05, $p = 0.0001$), whereas pretreatment with Y27632 partially blocked the contraction induced by thrombin (0.71 ± 0.01, $p = 0.0068$). Y27632 alone inhibited the spontaneous contraction of gel (1.06 ± 0.03) compared to PBS control ($p = 0.0001$). Further, Y-27632 inhibited thrombin-induced phosphorylation of MLC2 (Fig 4C). These data suggest that thrombin activates two kinases, MLCK and ROCK, and that both kinases phosphorylate MLC2.

We next tested whether thrombin-induced increases in MLC2 phosphorylation may involve decreased myosin phosphatase activity. Myosin phosphatase dephosphorylates MLC and inhibits activation of acto-myosin [27]. MYPT1, the targeting subunit of myosin

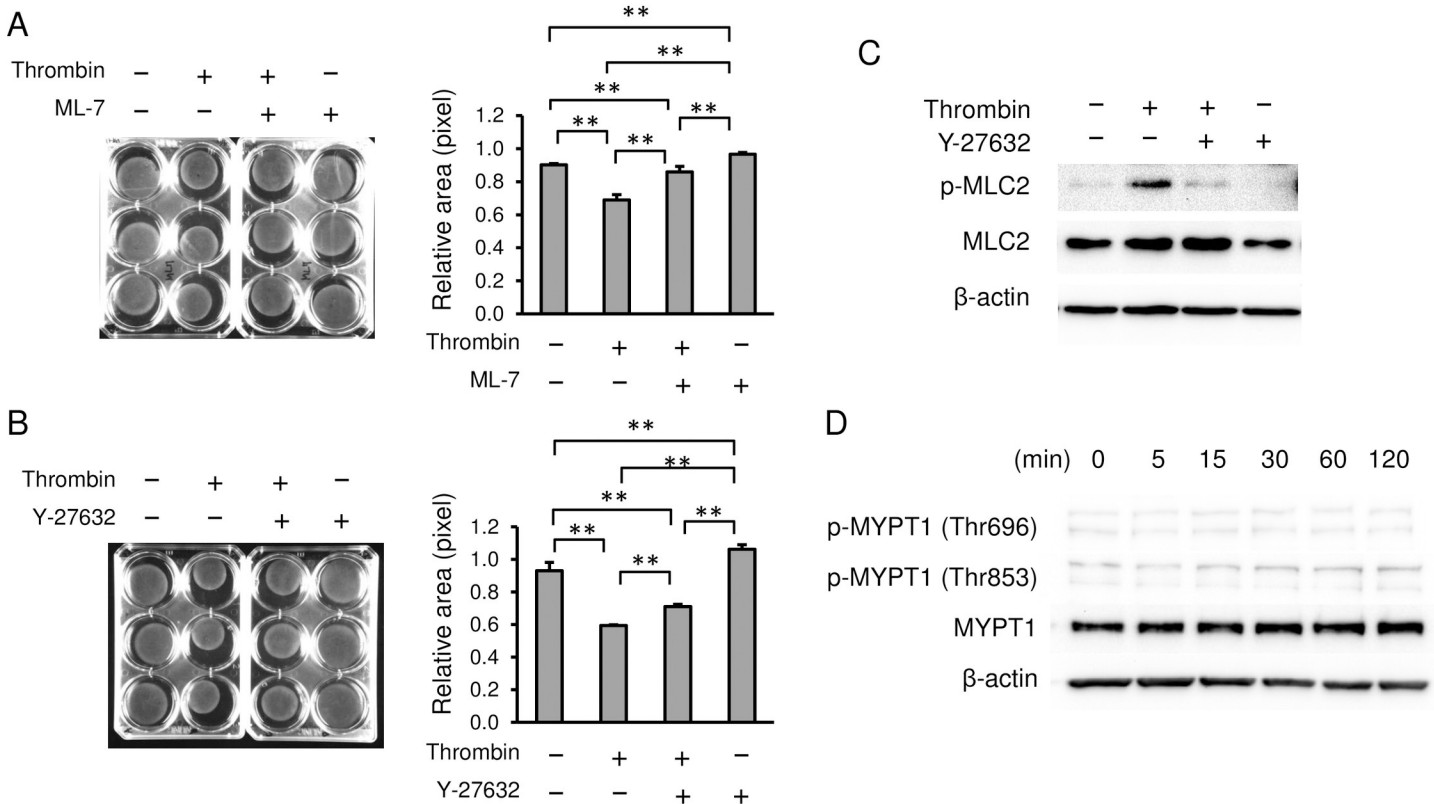

**Fig 4. Thrombin activates myosin by MLCK and ROCK. (A, B)** Collagen lattice assay with MLCK and ROCK inhibitors. Myometrial cells were pretreated with **(A)** 10 μM MLCK inhibitor ML7, or **(B)** 1 μM of Rho-kinase inhibitor Y-27632 for 1 h, and treated with 2 U/mL of thrombin for 30 min. Representative image (left panels) and quantification of gel areas (right graphs). n = 3 in each group. **(C)** Immunoblots of phosphorylated MLC2 (p-MLC2), MLC2, and β-actin of myometrial cells. **(D)** Immunoblots of phosphorylated MYPT1 (Thr696 and Thr853), MYPT1, and β-actin of myometrial cells. Myometrial cells were treated with 2 U/mL of thrombin as indicated time. **, $p < 0.01$. The experiments were repeated three times.

phosphatase, has two different activating and inhibitory phosphorylation sites. Phosphorylation of MYPT1 at sites Thr696 and Thr853 results in inhibition of myosin phosphatase and thereby increased MLC2 phosphorylation [28]. Thrombin, however, did not phosphorylate MYPT1 at these sites (Fig 4D), suggesting that thrombin-induced contractions are initiated through MLCK-induced activation of MLC2 phosphorylation but not inhibition of myosin phosphatase.

### Thrombin increased PTGS2 and PGF2α in myometrial cells

We next assessed the effect of thrombin on prostaglandin (PG) synthesis in myometrial cells. 1 U/mL of thrombin increased *prostaglandin-endoperoxidase 2* (*PTGS2 or cyclooxygenase 2*) mRNA 8-fold at 24 h with 10-fold increases using 4 U/mL of thrombin (Fig 5A). Thrombin increased *PTGS2* mRNA synthesis as early as 4 h, reaching plateau levels between 12 and 24 h (Fig 5B). PGF2α in the media of thrombin-treated myometrial cells increased dose- and time-dependently (Fig 5C and 5D). PGE2 in the media was also increased by thrombin, although the degree of increase was less than that for PGF2α (Fig 5E and 5F). Indomethacin, a PTGS2 inhibitor, partially blocked thrombin-induced myometrial contraction (Fig 5G and 5H). The relative pixel area with treatment of thrombin was decreased to 0.75 ± 0.05 compared to that of PBS (0.96 ± 0.03, $p = 0.0001$), whereas pretreatment with indomethacin blocked the contraction of gel by thrombin (0.87 ± 0.01, $p = 0.0020$). Indomethacin alone inhibited the

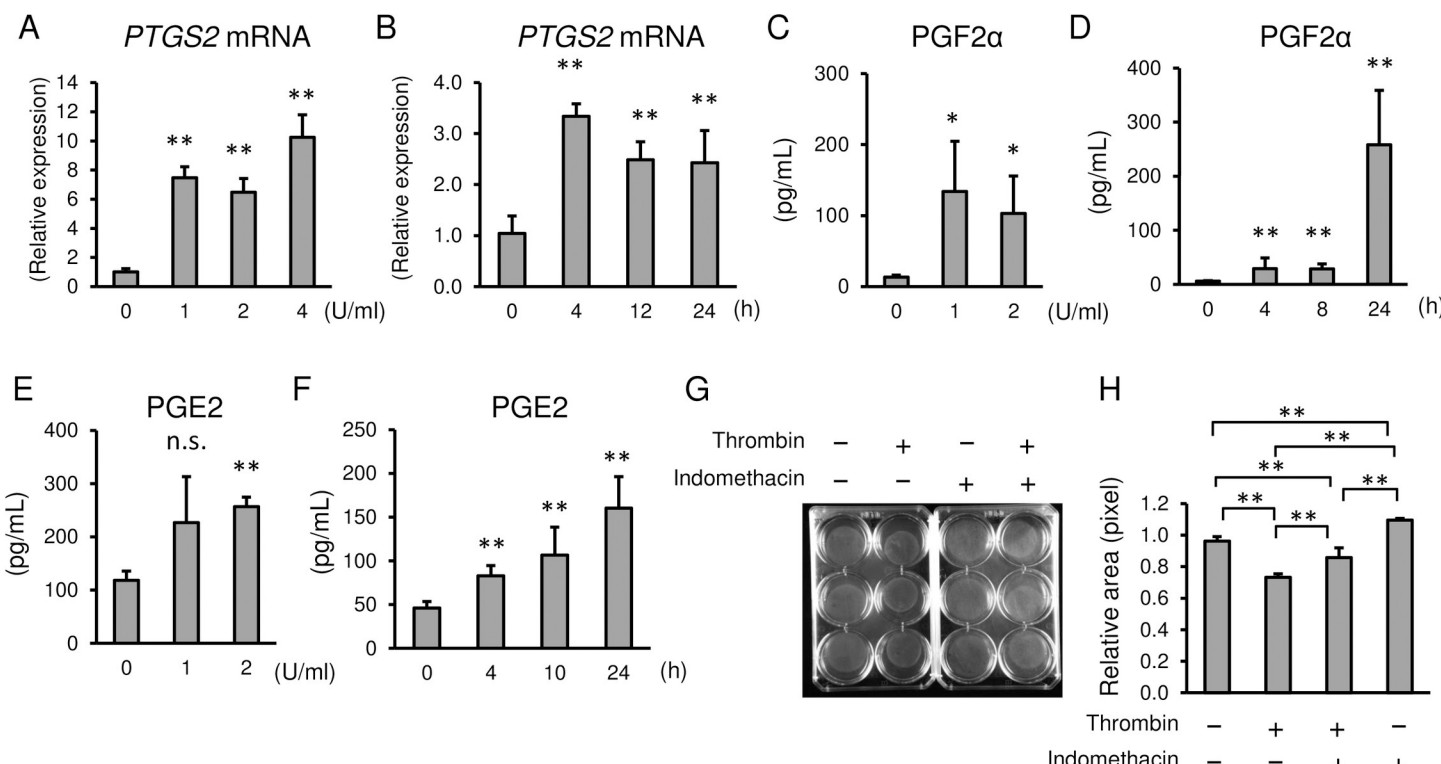

**Fig 5. Effect of thrombin on *PTGS2* mRNA expression and prostaglandin synthesis in myometrial cells. (A, B)** Thrombin-induced increases of *PTGS2* mRNA at 24 h with different doses of thrombin **(A)**, or 2 U/mL of thrombin as a function of time **(B)**. **(C-F)** Thrombin-induced increases of PGF2α **(C, D)** and PGE2 **(E, F)** in the media of thrombin-treated cells. Dose-dependent changes at 24 h **(C and E)** and time course with 2 U/mL of thrombin **(D, F)**. **(G and H)** Collagen lattice assay of myometrial cells at 30 min with 2 U/mL of thrombin, pretreated with 10 μM indomethacin for 4 h. **(G)** Representative image and **(H)** quantification of gel areas. n = 3 in each group. *, $p < 0.05$, and **, $p < 0.01$. The experiments were repeated three times.

spontaneous contraction of myometrial cells-embedded gel (1.11 ± 0.02) compared to PBS control ($p = 0.0283$). The data suggest that thrombin-induced contractions are mediated not only through direct thrombin-PAR1-MLC2 activation, but also by prostaglandin synthesis. The latter may lead to prolonged tonic contractions associated with placental abruption or massive subchorionic hematoma.

We next assessed the effect of thrombin on the expression of other contraction-associated proteins. Gene expression of prostaglandin E2 and F2α receptors (*PTGER1*, *PTGER3*, *and PTGFR*), oxytocin receptor (*OXTR*), and the gap junction protein connexin 43 (*GJA1*) were also analyzed (Fig 6A). Expression of *GJA1*, *OXTR*, and *PTGER1* mRNA was not altered by thrombin whereas mRNA levels of *PTGER3* and *PTGFR* were decreased (Fig 6A). Interestingly, mRNA expression of the inflammatory cytokine, IL-1β, was upregulated by thrombin (Fig 6A). Treatment of myometrial cells with IL-1β robustly increased *PTGS2* mRNA. PGE2, but not PGF2α, increased *IL1B* mRNA (Fig 6D). This suggests that PGE2 is increased by positive feedback through IL-1β (Fig 6F). Both thrombin and IL-1β decreased *PTGER3* mRNA (Fig 6A and 6B), and pretreatment of indomethacin alleviated the decrease of *PTGER3* mRNA (Fig 6C). In addition, both PGE2 and PGF2α decreased *PTGER3* mRNA (Fig 6D and 6E). Collectively, the thrombin-induced decrease of *PTGER3* mRNA was mediated by PGE2 and PGF2α through IL-1β (Fig 6F). In contrast, IL-1β did not change *PTGFR* mRNA (Fig 6B), and indomethacin did not recover the decrease of *PTGFR* mRNA by thrombin treatment (Fig 6C). Moreover, neither PGE2 nor PGF2α regulate *PTGFR* mRNA (Fig 6D and 6E). These findings

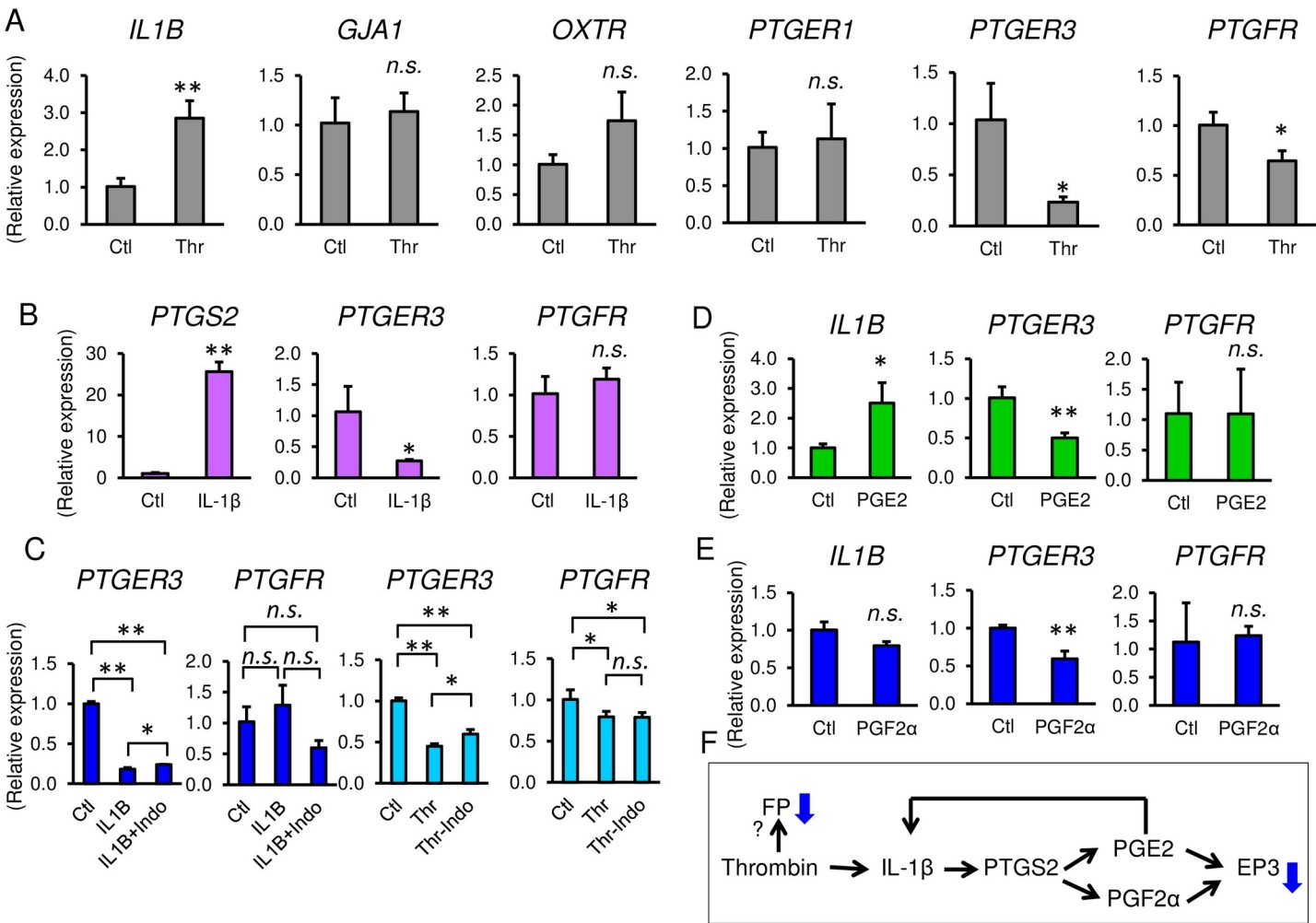

**Fig 6. Effect of thrombin on contraction-associated proteins in myometrial cells.** (**A**) *IL1B*, *GJA1*, *OXTR*, *PTGER1*, *PTGER3*, and *PTGFR* mRNA gene expression in thrombin (2 U/mL)-treated myometrial cells at 24 h. n = 3 in each group. (**B**) Gene expressions of myometrial cells with treatment of 0.1 ng/mL of IL-1β at 24 h. (**C**) mRNA expressions of *PTGER* and *PTGFR* with treatment of thrombin or IL-1β, pretreated with indomethacin. (**D**) Gene expressions of myometrial cells with treatment of 0.1 μM of PGE2 (**D**) or 0.1 μM of PGF2α (**E**) at 24 h. n = 3. *, $p < 0.05$, and **, $p < 0.01$. (**F**) Scheme of regulation of contraction associated proteins by thrombin. The experiments were repeated three times.

suggest that the thrombin-induced decrease of *PTGFR* mRNA might be regulated by molecules other than prostaglandins.

## Progesterone inhibited thrombin-induced myometrial contraction

Finally, to test the therapeutic potential of progesterone for thrombin-induced myometrial contraction, we used collagen lattice assays (Fig 7A). Progesterone partially blocked thrombin-induced myometrial contractions at two physiologic concentrations (1 μM, Fig 6A and 0.1 μM, S1 Fig). Under pretreatment with 1 μM of progesterone, the relative pixel area by thrombin treatment was decreased to 0.74 ± 0.03 compared to that of PBS (0.93 ± 0.01, $p = 0.0001$), whereas pretreatment with progesterone partially blocked the contraction by thrombin (0.84 ± 0.02, $p = 0.0010$). Progesterone alone relaxed the myometrial cells (1.02 ± 0.01) compared to PBS control ($p = 0.0030$). In addition, progesterone treatment resulted in inhibition of thrombin-induced increases of *PTGS2* and *IL1B* mRNA (Fig 7B).

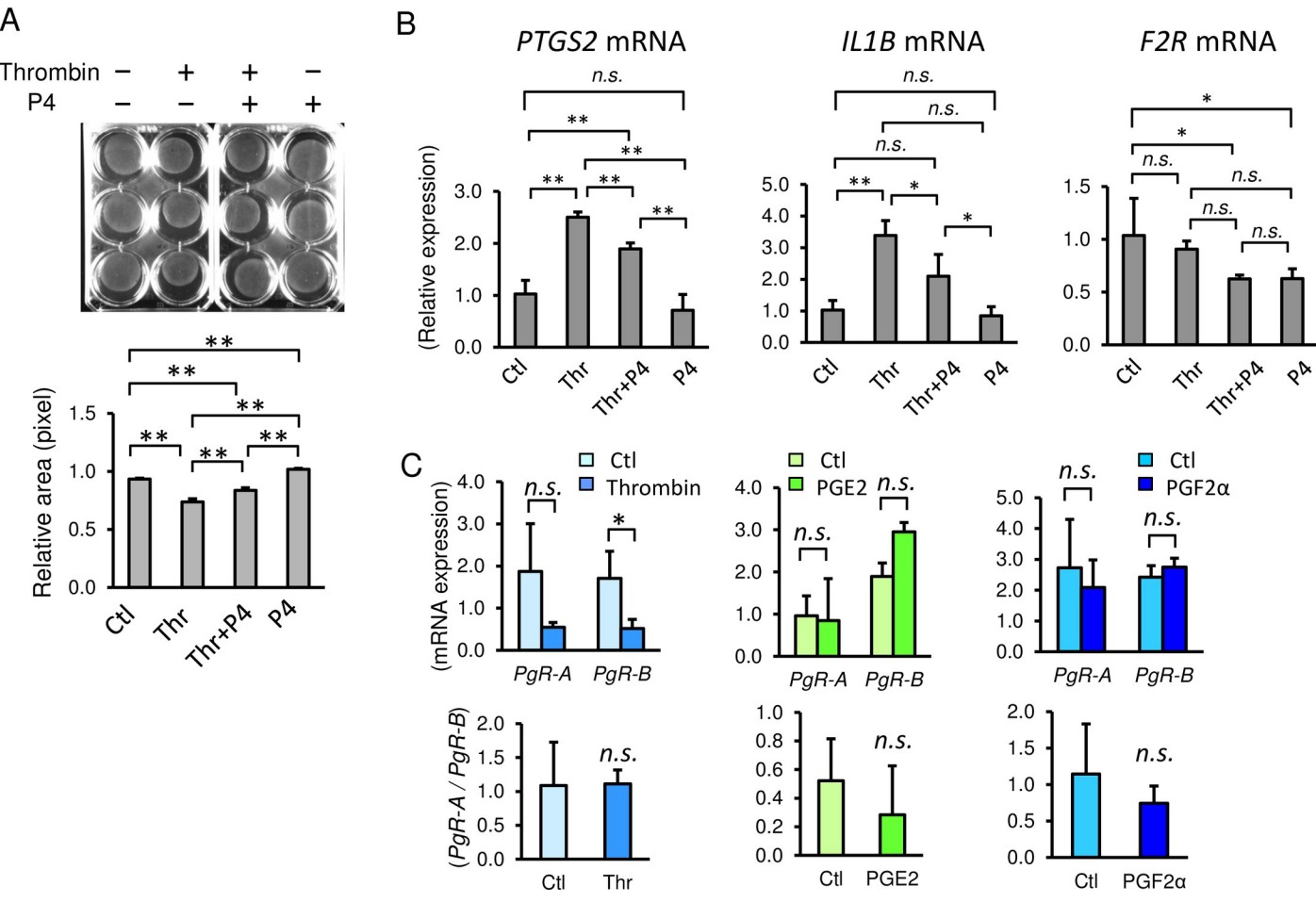

**Fig 7. Effect of progesterone (P4) on thrombin-induced myometrial cell contractions.** (A) Collagen lattice assay of myometrial cells at 30 min with 2 U/mL of thrombin (Thr) pretreated with 1 μM progesterone (P4) for 1 h. Representative image (upper panel) and quantification of gel areas (lower graph). (B) Inhibition of thrombin-induced increases of *PTGS2*, *IL1B*, and *F2R* mRNA by P4. Myometrial cells were pretreated with 1 μM of P4 for 1 h, and then treated with 2 U/mL of thrombin. (C) Gene expressions of progesterone receptor-A and–B (*PgR-A* and *PgR-B*) with 24 h treatment of 1 U/mL of thrombin, 10 nM of PGE2, and 10 nM of PGF2α (upper graphs) and *PgR-A* to *PgR-B* ratio (lower graphs). n = 3 in each group. *, $p < 0.05$, and **, $p < 0.01$. The experiments were repeated three times.

mRNA level of thrombin receptor, PAR1 gene, (*F2R*) was not changed by thrombin treatment, but expression of *F2R* mRNA was decreased by progesterone, notwithstanding the presence of thrombin (Fig 7B). Collectively, the results indicate that progesterone partially alleviated thrombin-induced myometrial contractions.

Progesterone receptors (PRs) control progesterone responsiveness. Human PR has two major isoforms. PR-B is the full-length progesterone receptor and PR-A is the truncated form of PR-B [29]. PR-B principally mediates relaxatory actions of progesterone, whereas PR-A inhibits the transcriptional activity of PR-B. Therefore, the PR-A to PR-B ratio regulates the transcriptional activity of progesterone. The effect of thrombin on the mRNA expression of progesterone receptors (*PgR-A* and *PgR-B*) was analyzed in primary myometrial cells. The abundance of total PR (*PgR-A* plus *PgR-B*) was decreased by thrombin (Fig 7C). The *PgR-B* mRNA level was particularly significantly decreased by thrombin (Fig 7C). Consequently, the *PgR-A* to *PgR-B* ratio was not changed by thrombin (Fig 7C). In contrast, treatment of PGE2 or PGF2α did not change *PgR-A* and *PgR-B* mRNA levels, and the *PgR-A* / *PgR-B* ratio was not changed (Fig 7C). Therefore, the suppressed transcription of PR seems to be mediated by

molecules other than prostaglandins. These data suggest that thrombin also strengthens myometrial contraction by reducing the expression of total abundance of progesterone receptors, although it was not regulated by prostaglandins.

## Discussion

It is empirically known that intrauterine bleeding such as decidual hemorrhage (subchorionic hematoma) or placental abruption causes uterine contraction, which sometimes leads to preterm birth. We previously showed that thrombin injection into pregnant mice caused preterm birth, suggesting a pathological role of thrombin *in vivo* [12]. Here, we further clarified the molecular mechanisms by which thrombin stimulates uterine contraction, and the potential of progesterone to antagonize thrombin.

In this study, we utilized uterus tissue from non-pregnant women because uterus tissue from pregnant women is rarely available in our facility. We tested the contraction of myometrial cells from a rare case of cesarean hysterectomy and found that the contraction of myometrial cells from pregnant uterus was similar to that from non-pregnant uterus. In addition, thrombin receptor PAR1 was expressed in both pregnant and non-pregnant myometrium (Fig 1). Previously, O'Sullivan et al. showed that thrombin and PAR1 activating peptide exerted a stimulatory effect on uterine contractions in both pregnant and non-pregnant myometrial tissues [30]. They also confirmed that there was no significant difference in sensitivity to thrombin between pregnant and non-pregnant myometrium. Therefore, our data obtained from non-pregnant myometrium provide valid insight into the mechanisms of preterm birth induced by intrauterine bleeding.

PAR1 was highly expressed in human pregnant myometrium, and thrombin and PAR1 activating peptide induced significant myometrial contraction, as previously reported [13, 14, 30]. PAR1 was originally identified as the thrombin receptor on platelets, and activation of PAR1 signaling stimulates platelet aggregation [31]. Thereafter, PAR1 activity was found to be associated with the hemostasis and thrombosis of platelets, vascular tone and permeability in the endothelium, and contraction and atherosclerosis of vascular smooth muscle cells [7, 32]. PAR1 is also involved in inflammation and tumor metastasis. O'Brien et al. showed that PAR1 (F2R) was expressed in both pregnant and non-pregnant myometrium, which is compatible with our data from immunofluorescence; they also showed that expression is higher during pregnancy compared to non-pregnancy [33]. Interestingly, PAR1 expression increased 9-fold during labor in human myometrium compared to the state of not in labor, indicating that the sensitivity to thrombin is increased during labor. The increased expression of PAR1 would contribute to stronger myometrial contraction during labor and puerperium.

Our data showed that the human uterus possesses a mechanism to sense abnormal intrauterine bleeding, and once bleeding occurs in emergency situations such as placental abruption, traumatic hemorrhage, or threatened abortion, myometrium contractions are initiated to expel the conceptus and protect the mother. PAR1 signal is then passed to myosin light chain via two kinases: MLCK and ROCK. Activation of G protein-coupled receptor, PAR1, by thrombin increases intracellular free $Ca^{2+}$ [13]. $Ca^{2+}$ binds to calmodulin, and the $Ca^{2+}$-calmodulin complex associates with the catalytic subunit of MLCK, which phosphorylates serine at position 19 on the regulatory light chain of MLC2 [17]. In contrast, upon activation of G-protein coupled receptor PAR1, Rho proteins are translocated to the cell membrane where guanine nucleotide exchange factors (Rho-GEFs) promote exchange of GDP to GTP of Rho. GTP-bound "turned-on" Rho proteins activate ROCK, which also phosphorylates the regulatory light chain of MLC2 [34]. In addition to direct activation of the actomyosin complex, we found that thrombin also increased expression of *PTGS2* mRNA thereby releasing PGE2 and

PGF2α from myometrial cells. This mechanism is similar to that in fetal membranes in which thrombin increases *PTGS2* mRNA and PGE2 synthesis of amnion *in vitro* and *in vivo* [12]. Hence, thrombin-stimulated prostaglandin synthesis of myometrium strengthens myometrial contractions in an autocrine fashion.

In human pregnancy, the placenta is the source of high progesterone concentrations at the maternal–fetal interface. Interruption of this communication may be highly localized in the case of placental separation/abruption and not reflected in maternal blood. Here, we showed that progesterone inhibited thrombin-induced *PTGS2*, *IL1B*, and *F2R* mRNA. All of these suppressions of prostaglandin synthesis, inflammatory cytokines, and expression of thrombin receptor would contribute to antagonizing myometrial contractions by thrombin. It is possible, therefore, that progesterone may serve a therapeutic potential in cases of premature uterine contractions caused by decidual hemorrhage. Relaxation of the myometrium may restore blood flow to the fetus and facilitate prolongation of pregnancy. We emphasize, however, that our experiments were conducted *in vitro* in the absence of blood vessels and a rich capillary bed. Thus, more research is warranted regarding the safety and efficacy of progesterone for intrauterine bleeding before it can be recommended.

Progesterone receptors regulate the contraction of the human uterus during pregnancy, and PR-A and PR-B expression is altered in myometrium in labor, increasing the ratio of the PR-A to PR-B level [35–37]. In our study, thrombin decreased the total expression of progesterone receptor, although the *PgR-A* to *PgR-B* mRNA ratio was not changed, and prostaglandins did not regulate the downregulation of PRs. Madsen et al. showed that the *PgR-A* / *PgR-B* mRNA ratio was increased by PGE2 and PGF2α in an immortal pregnant human myometrial cell line [35]. However, they also showed that the *PgR-A* / *PgR-B* mRNA ratio tended to return to the basal level when prostaglandin concentrations increased. We utilized a comparably high dose of prostaglandins (10 nM) in our experiments, so this might be why the *PgR-A* to *PgR-B* mRNA ratio was not altered in our study. In addition to the direct effect of myometrial contraction, thrombin further strengthens contractions by reducing the progesterone responsiveness caused by the downregulation of progesterone receptor expression. Further study is necessary to investigate the regulation of progesterone receptors by thrombin.

The ultimate purpose of this study is to provide basic information regarding the mechanisms of preterm labor and poor pregnancy outcomes of women with intrauterine bleeding (decidual hemorrhage). Here, we found that progesterone was useful for blockade of thrombin-induced contraction. This finding suggests the potency of progesterone in subchorionic hematoma of the first and second trimester to prevent preterm birth and spontaneous abortion, especially if the extraordinarily high levels of progesterone at the placental–myometrial interface are disrupted by bleeding.

Thrombin–antithrombin complex (TAT) levels increases during normal pregnancy [38]. The mean TAT level is 1.9 μg/L in non-pregnant women regardless of menstrual cycle phase, but increases to 16.0 ± 2.8 μg/L in the 2nd trimester, 21.5 ± 11.9 μg/L at term, and 28.6 ± 12.8 μg/L in the 2nd stage of labor [15]. In addition, TAT further increases in the 3rd stage of labor upon separation of the placenta [15, 16]. This is due to the release of thromboplastic substances at the site of placental separation. It seems that released thrombin from the placenta-attached site to the uterus facilitates the contraction of myometrium (uterine involution), which prevents massive hemorrhage at parturition. Therefore, the abnormal increase of thrombin mid-trimester is a pathological process that leads to preterm birth, whereas the increase in maternal circulating thrombin in the 3rd stage of labor and parturition plays a role in the physiological contraction of myometrium.

Recently, it has been suggested that thrombin may be produced from fetal membrane not only from intrauterine bleeding but also by bacterial infection [39]. This finding is consistent

with our previous data in which the protease activity of thrombin was significantly increased in amnion from women with preterm birth without intrauterine bleeding [12]. Therefore, inhibition of thrombin signaling pathways has potential to treat labor that is not associated with intrauterine bleeding.

## Conclusions

Thrombin induces myometrial contractions through both direct (PAR1-mediated myosin activation) and indirect (PTGS2-mediated increases in prostaglandin synthesis) mechanisms. The therapeutic potential of progesterone was suggested for preterm labor complicated by intrauterine bleeding that is not life-threatening.

## Supporting information

**S1 Fig. Expression of protease-activated receptor 1 (PAR1) and thrombin (F2) in human pregnant myometrium.**
(TIF)

**S2 Fig. Thrombin increased contraction of primary human myometrial cells from pregnant uterus.**
(TIF)

**S3 Fig. Original uncropped and unadjusted images of immunoblots of Fig 3B.**
(TIF)

**S4 Fig. Original uncropped and unadjusted images of immunoblots of Figs 3C and 4C.**
(TIF)

**S5 Fig. Original uncropped and unadjusted images of immunoblots of Fig 4D.**
(TIF)

**S1 Movie. Time-lapse live imaging of thrombin-treated myometrium.** Available at https://doi.org/10.5281/zenodo.3240679.
(TXT)

## Acknowledgments

We thank Professor R. Ann Word (University of Texas, Southwestern Medical Center, Department of Obstetrics and Gynecology) for critical comments on this manuscript. We thank Ms. Mizuho Ohshima and Ms. Ayako Yoshida for technical assistance, and Ms. Iku Sugiyama and Ms. Akiko Abe for editorial assistance. We also thank the shared resource cores in the Medical Research Support Center for confocal microscopy at Kyoto University Graduate School of Medicine.

## Author Contributions

**Conceptualization:** Haruta Mogami.

**Formal analysis:** Haruta Mogami.

**Funding acquisition:** Haruta Mogami.

**Investigation:** Fumitomo Nishimura, Haruta Mogami, Kaori Moriuchi.

**Methodology:** Haruta Mogami.

**Project administration:** Haruta Mogami.

**Supervision:** Haruta Mogami, Yoshitsugu Chigusa, Masaki Mandai, Eiji Kondoh.

**Writing – original draft:** Fumitomo Nishimura, Haruta Mogami.

**Writing – review & editing:** Haruta Mogami.

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
