## [Decision Letter · Decision Letter 0]

14 Oct 2019

PONE-D-19-21553

Mechanisms of Thrombin-Induced Myometrial Contractions: Potential Targets of Progesterone

PLOS ONE

Dear Dr. Mogami,

Thank you for submitting your manuscript to PLOS ONE. After careful consideration, we feel that it has merit but does not fully meet PLOS ONE’s publication criteria as it currently stands. Therefore, we invite you to submit a revised version of the manuscript that addresses the points raised during the review process.

The reviewers agree that the manuscript tackles an important and under-researched topic. The most critical issues to be addressed for possible publication are clarification of the number of biological vs. technical replicates presented and included in the statistical analyses and also to make available all data that form the basis of the manuscript. A minimum of 3 (three) biological replicates is mandatory in all experiments. The conclusions and interpretations should be revised to take into account the comments put forward by the expert reviewers. The Authors should seriously consider to add experiments along the suggestions by Reviewer 2. 

We would appreciate receiving your revised manuscript by Nov 28 2019 11:59PM. To enhance the reproducibility of your results, we recommend that if applicable you deposit your laboratory protocols in protocols.io, where a protocol can be assigned its own identifier (DOI) such that it can be cited independently in the future. For instructions see: http://journals.plos.org/plosone/s/submission-guidelines#loc-laboratory-protocols

We look forward to receiving your revised manuscript.

Kind regards,

Tamas Zakar

Academic Editor

PLOS ONE

**Journal Requirements:**

https://link.springer.com/book/10.1007%2F978-981-10-2489-4

The text that needs to be addressed is in the Introduction section.

In your revision ensure you cite all your sources (including your own works), and quote or rephrase any duplicated text outside the methods section. Further consideration is dependent on these concerns being addressed. "

3. Thank you for including your ethics statement: All human myometrial tissues were obtained in accordance with the ethics committee of the Graduate School of Medicine, Kyoto University after written consent (G1149).

**Comments to the Author**

1. Is the manuscript technically sound, and do the data support the conclusions?

Reviewer #1: No

Reviewer #2: Yes

2. Has the statistical analysis been performed appropriately and rigorously? 

Reviewer #1: No

Reviewer #2: Yes

3. Have the authors made all data underlying the findings in their manuscript fully available?

Reviewer #1: No

Reviewer #2: Yes

4. Is the manuscript presented in an intelligible fashion and written in standard English?

Reviewer #1: Yes

Reviewer #2: Yes

5. Review Comments to the Author

Reviewer #1: IS THE MANUSCRIPT TECHNICALLY SOUND, AND DO THE DATA SUPPORT THE CONCLUSIONS?

The manuscript follows a logical approach of using the collagen lattice assay to measure contractility induced in primary myometrial cells following treatment with thrombin. The extent of contraction is quantitated by measuring the area of lattice using Image J software.

Figure 1 demonstrates that both thrombin and its receptor, PAR1, are present in pregnant and non-pregnant human myometrium.

Figure 2 demonstrates that thrombin induces contraction of primary myometrial cells, and that activation of PAR1, by means of the activating peptide, also induces contraction. Furthermore, the PAR1 inhibitor, SCH79797, inhibited the vast majority (but not all) of the thrombin-induced contraction.

Curiously, page 9 line 7 indicates that the primary myometrial cells utilised to generate the data in Figure 2 were derived from non-pregnant myometrium, rather than pregnant myometrium. Is this correct or should this say

Figure 3 demonstrates that thrombin induces MLC-ser19 phosphorylation, and that this phosphorylation is inhibited by the PAR1 inhibitor. Densitometric

Figure 4 provides evidence that thrombin-induced contraction of primary myometrial cells is mediated by myosin light chain kinase and ROCK, as evidenced by the MLCK inhibitor (ML-7) and the ROCK inhibitor (Y-27632) both partially inhibiting thrombin-induced MLC-ser19 phosphorylation. Additionally, Panel D demonstrates that thrombin does not promote myometrial contraction by inhibiting (phosphorylating) the regulatory subunit of myosin phosphatase, MYPT1.

Figure 5 demonstrates that thrombin-induced contraction of primary myometrial cells is associated with (i) significantly upregulated expression of PTGS2 (COX2) after 4 h, (ii) significantly increased production of PGF2 alpha after 4 h, with maximum levels observed after 24 h, (iii) significantly upregulated expression of IL1B, (iv) significantly down-regulated expression of both PTGER3 (EP3) and PTGFR (FP), and (v) no change in expression of GJA1 (CX43), OXTR or PTGER1 (EP1). Additionally, panels E and F demonstrate that indomethacin partially inhibits the thrombin-induced contraction, demonstrating that thrombin-induced contraction is in-part indirect via stimulating prostaglandin production.

Figure 6 demonstrates that thrombin-induced contraction of primary myometrial cells is partially inhibited by pre-treating the cells with progesterone (P4) or medroxyprogesterone acetate (MPA).

The conclusions drawn are appropriate for the data, however, questions remain as to the whether the data is comprised of technical replicates or biological replicates.

HAS THE STATISTICAL ANALYSIS BEEN PERFORMED APPROPRIATELY AND RIGOROUSLY?

Further explanation is required as to what is being considered a replicate. For instance, the Figure 2 legend says “n=3 in each group. Experiment was repeated with cells from another preparation with identical results”. This reads as though the study was performed with 3 technical replicates (ie myocytes from 1 woman plated across multiple wells and then the treatment applied to 3 of those wells), and then repeated in only one other woman. If so, this would mean that only n=2 biological replicates have been examined (with 3 technical replicates for each biological replicate). The minimum requirement is n=3 biological replicates, so if n=3 biological replicates have indeed been performed, this needs to be explained clearly in the materials and methods.

HAVE THE AUTHORS MADE ALL DATA UNDERLYING THE FINDINGS IN THEIR MANUSCRIPT FULLY AVAILABLE?

Uncropped versions of the western blots have been made available in the supplementary data, however, only it appears that data may only be available for n=1 biological replicate. Additionally, all the biological replicates for the collagen lattice assays (minimum required is n=3) do not appear to be available.

IS THE MANUSCRIPT PRESENTED IN AN INTELLIGIBLE FASHION AND WRITTEN IN STANDARD ENGLISH?

English quality is quite good, however, there are some minor grammatical errors that should be addressed. Some of these issues are outlined below, but additional proof-reading is required with attention to the correct placement of commas.

Requested changes and corrections:

Introduction:

1. There is no detail provided in relation to the Image J measurement. Although this is a standard

Materials and Methods:

2. Page 5, line 23: “and then added thrombin” should be “and then thrombin was added”

3. Correct SI unit for litre is ‘L’, not ‘l’. Apply throughout entire manuscript, including figures.

4. There is no detail provided in relation to the Image J measurement. Although this is a standard assay, additional information as to how the Image J analysis was performed would be welcomed to explain the result. For instance, in most graphs showing the results of the collagen assays, the mean relative pixel area is not 1.0, which indicates that the collagen discs did not occupy 100% of the area measured. Presumably the area measured was the size of the well? Such information should be conveyed as it affects the interpretation of the results (i.e. the extent of thrombin-induced contraction is not a percentage of the pixel area of the untreated control).

5. It should be possible to fully understand the experiments conducted and the results obtained without having to refer to the figures. As such, in the materials and methods, please provide the concentrations of treatments in-text (for example, page 5, line 21).

6. Page 5, line 21: provide O2 and CO2 percentages.

7. There is no mention of protein quantitation method and no details provided for SDS-PAGE, such as the amount of protein loaded per lane (supplementary figures indicate that 20 ug per lane was loaded, however, this information should be provided in-text in the methods). Type of membrane used for western blotting not indicated (PVDF or nitrocellulose?).

8. Antibody dilutions are provided, however, this does not provide any indication of the quantity of antibody used, as different antibodies come at different dilutions. While it is common practice to merely report antibody dilutions, the authors are encouraged to additionally report the actual final antibody concentration utilised (i.e. 0.1 ug/mL in 20 mL of TBST). Not essential but encouraged.

Results:

9. In the results, the collagen lattice assays graphs convey the reduction in relative pixel area, however, nowhere in the manuscript are the actual quantitated values provided. For each of the collagen lattice assay graphs, please report the mean pixel area (with +/- SD) for the different treatments in-text with the results. Additionally, the figure legends indicate that “*, p < 0.05, and **, p <0.01”, however, please provide the p-values in-text during the results. For example: Thrombin treatment significantly decreased relative pixel area compared to control (0.95 +/- 0.02 versus 0.6 +/- 0.03; p=0.03).

10. In the legend for Figure 5, Section C states “2U/ml of thrombin for 12 h”, however, the X-axis of panel C (which displays the effect of thrombin on PGF2 alpha levels) appears to show thrombin treatments at 1 and 2 U/mL.

11. Page 8, line 23-24: “Although not observed in the center of hemorrhage, PAR1 was expressed in myometrial cells peripheral to the bleeding site”. FYI: This is an interesting observation that leaves me curious as to why that may be.

12. Page11, line 10: ‘media of thrombin-treated…’ instead of ‘media by thrombin-treated…’

13. Number of replicates are not adequately identified throughout the results, either in -text or in the figure legends. For example, the legends for Figures 3 and 4 do not indicate how many biological replicates were performed for the western blots (i.e. was only 1 western blot performed for 1 woman/biological replicate, or was the blotting performed for primary myocytes isolated from n=3 different women?) or immunocytochemistry.

14. Page21, Figure 2 legend: Please replace “identical results” with “consistent results”, as it is impossible that identical results were observed across replicates.

General:

15. The throughout the entire manuscript and figures, authors are encouraged to use the correct protein names; i.e. Prostaglandin-Endoperoxide Synthase 2 (gene name: ‘PTGS2’), rather than cyclooxygenase (‘COX2’), with gene names in all-caps and non-italicised.

16. The throughout the entire manuscript and figures, the authors are encouraged to use the correct gene names (presented in app-caps and italicised). COX2 = PTGS2, CX43 = GJA1, EP1 = PTGER1, EP3 = PTGER3, FP = PTGFR. Refer to genecards.org.

17. Grammar: Use ‘protease-activated receptor 1’, not ‘protease-activated receptor-1’.

18. ‘PAR-1’ and ‘PAR1’ used interchangeably. Additionally, ‘protease-activated receptor 1 is firdst used in the abstract and the abbreviation (PAR1) is provided. In-text, however, the PAR1 abbreviation is used without providing the full name (page 5, line 2).

19. Page 14, line 12: what is meant by “activity”? This is vague and needs clarification.

20. Page 14, line 13-15: “Therefore, inhibition of thrombin signaling pathways has potential to treat preterm labor associated “without” intrauterine bleeding.”. Please correct this sentence: “Therefore, inhibition of thrombin signaling pathways has potential to treat labor that is not associated with intrauterine bleeding”.

Comments:

The primary myometrial cells utilised were isolated from myometrium obtained from non-pregnant premenopausal women. The manuscript would benefit from authors commencing the Discussion by justifying their case for not using pregnant myometrium and provide a brief argument as to why findings gleaned from nonpregnant primary uterine myocytes still provide valid insight into the mechanisms underpinning premature birth.

There is no mention of existing literature that examines the role/regulation of thrombin in normal labour. Ie. What happens to thrombin levels during normal labour? These studies should be briefly mentioned as existing evidence indicates that thrombin activity rises during normal, uncomplicated labour, suggesting that thrombin activity is a normal component labour. Additionally, this has implications for the author’s claim that (page 3, line 20) “Collectively, these data suggest that thrombin is a pathogenic factor in initiation of preterm birth”, which suggests a causative role in preterm birth. It is possible that thrombin activity in indeed causative of premature labour, as supported by thrombin by inducing PTB in mice, but it is also possible that thrombin activity rises as a result of the woman being in labour, whether the labour is at term or preterm. As such, the manuscript would benefit by highlighting that the key underlying issue is that dysregulation of thrombin activity has the potential to cause the premature onset of labour, leading to preterm birth.

There is no discussion on how expression of the genes encoding thrombin (F2) and PAR1 (F2R) are regulated in the myometrium. The manuscript would benefit from discussing what is known about F2 (primarily) and F2R (secondly) expression in the myometrium (and other tissues), and how dysregulation of expression of these genes could be causal to preterm birth.

The authors suggest a potential clinical role for progesterone in inhibiting preterm labour complicated by intrauterine bleeding, based on the ability of progesterone to partially inhibit thrombin-induced myometrial contraction. Existing literature indicates that the relaxatory effects of progesterone are mediated by PR-B, but also that prostaglandins increase the PR-A/PR-B ratio. Given that the authors present data indicating that thrombin induces production of prostaglandins, the manuscript would be benefit from discussing how thombin-induced prostaglandin production may affect the PR-A/PR-B ratio, and how this may affect the clinical usefulness of progesterone therapy.

I’m not sure whether it is an artefact of the production of the draft version of the manuscript, but the figures appear to be low resolution. There is extensive pixilation. Hopefully the final version will have figures of greater clarity.

Overall, this study presents some interesting findings that, if true, will make a valuable contribution to advancing our understanding of the mechanisms contributing to the syndrome of preterm birth. However, at this stage, there appears to be insufficient biological replicates, with the results instead based on the statistical analysis of technical replicates (if this reviewer’s interpretation of the data is incorrect, then could the authors please provide some clarity). If the findings persist following the incorporation of additional biological replicates, the study will make a valuable contribution to the field.

Reviewer #2: This interesting paper explores the ability of thrombin, released most often during intrauterine hemorrhage, on its ability to stimulate other mediators (IL-1beta, prostaglandins, COX-2) via the PAR-1 receptor and MLCK intracellular pathways and the ability of progesterone to block these

The writing of the manuscript, techniques, data analysis and figures are very good. I found just one typo in the MS ('as' should be replaced with 'at' in one figure legend).

My major recommendation though is that the there are uncompleted experiments that will improve the information shared and conclusions reached. These begin by noting that thrombin increased COX-2 expression and PGF2alpha output, but that expression of the FP receptor decreased. The authors did not address this important observation. I am quite certain the increased PGF2a produced cause the down regulation of its receptor, FP. Although PGE2 was not assayed, I'm also certain that thrombin increased it, too, and its higher concentrations led to the down regulation of the EP3 receptor noted in the figures. Again, this was not addressed very well by the authors.

I am suggesting that the authors perform a few more simple experiments to address these interesting observations in order to explain their results better. One of the questions that derives, is does thrombin stimulate a decrease in the expression of EP3 and FP directly or is this due to increased prostaglandins that down regulate their receptors. Thus if indomethacin or a COX-2 specific inhibitor is co-administered with thrombin, what is the outcome with the PG receptors? I would suggest that in this case, there is no down-regulation of EP3 or FP. If true, then administration of PGE2 or PGF2a and assessing expression of EP3 and FP should confirm that PGs down regulate their receptor expression. I propose that thrombin increases IL-1b and that IL-1b increases expression of EP3 and FP as well as COX-2. The reason that thrombin stimulates contractions is that the action of PGE2 and PGF2a is not lost (due to large down regulation of their receptors) because IL1b is stimulating their up-regulation. Hence they are present enough to respond to their agonists and effect contraction.

However, it is noted from the data that thrombin administration also increases IL-1b. But IL-1b on its own can stimulate COX-2 expression and it increases the expression of FP (see Mol Hum Reprod. 2015 Jul;21(7):603-14.). These should be confirmed in these cells. If true then the question derives how can thrombin stimulate myometrial contraction if it leads to the down regulation of FP or EP3 as shown by the limited data presented? In this case, stimulate cells with IL-1b in the absence or presence of indomethacin or a COX-2 inhibitor. I suspect there will be a large increase in EP3 and FP expression when COX-2 is inhibited, and less or even a decrease in expression when COX-2 activity is intact.

Finally, one begins to wonder where the action of progesterone or MPA is at. Is it directed at inhibiting COX-2 action directly, or is it directed at inhibiting expression of IL-1beta or even directed at PAR-1 expression? I strongly suggest to the investigators and to the Editor to include these important experiments in this manuscript to explain more completely the relationships hinted at by the data that is presented.

6. PLOS authors have the option to publish the peer review history of their article (what does this mean?). If published, this will include your full peer review and any attached files.

Reviewer #1: No

Reviewer #2: No

---

## [Author Response · Author response to Decision Letter 0]

25 Feb 2020

Response to the editor and reviewers

PONE-D-19-21553

Mechanisms of Thrombin-Induced Myometrial Contractions: Potential Targets of Progesterone

PLOS ONE

Dear Dr. Mogami,

Thank you for submitting your manuscript to PLOS ONE. After careful consideration, we feel that it has merit but does not fully meet PLOS ONE’s publication criteria as it currently stands. Therefore, we invite you to submit a revised version of the manuscript that addresses the points raised during the review process.

The reviewers agree that the manuscript tackles an important and under-researched topic. The most critical issues to be addressed for possible publication are clarification of the number of biological vs. technical replicates presented and included in the statistical analyses and also to make available all data that form the basis of the manuscript. A minimum of 3 (three) biological replicates is mandatory in all experiments. The conclusions and interpretations should be revised to take into account the comments put forward by the expert reviewers. The Authors should seriously consider to add experiments along the suggestions by Reviewer 2. 

Tamas Zakar

Academic Editor

PLOS ONE

Dear Dr. Zakar,

Thank you very much for your response in regard to our submitted manuscript, “Mechanisms of thrombin-induced myometrial contractions: Potential targets of progesterone” (PONE-D-19-21553). We are most grateful to you and the expert reviewers for the very valuable and constructive comments and suggestions. We have revised the manuscript accordingly and are confident that it has been much improved. We would like to resubmit it for publication in your esteemed journal. Below, please find our detailed responses to each of the comments. 

All of the experiments were repeated at least three times to meet the biological replicates requirements, and the results were regenerated and shown in the Figures of Replicated Experiments. Although progesterone clearly blocked thrombin-induced myometrial contraction, we were unable to obtain consistent results in our MPA experiments. As such, we removed the MPA data from our manuscript (Fig. 6C and 6D). We are confident that natural progesterone (P4) has a protective effect against thrombin-induced myometrial contraction. 

The additional experiments suggested by Reviewer 2 were performed and added to the revised manuscript. We made all of the data available in the manuscript and Supporting information files.

Our responses are written in blue. All revisions in the manuscript are also written in blue. 

Journal Requirements:

>>> Thank you for bringing this to our attention. We have revised our manuscript to follow PLOS ONE’s style requirements. The file naming has also been corrected. 

https://link.springer.com/book/10.1007%2F978-981-10-2489-4

The text that needs to be addressed is in the Introduction section.

In your revision ensure you cite all your sources (including your own works), and quote or rephrase any duplicated text outside the methods section. Further consideration is dependent on these concerns being addressed. "

>>> We cited the book “Precision Medicine in Gynecology and Obstetrics” in the references. This book contains our review article on thrombin and preterm birth. The sentence was modified in the revised manuscript in order to avoid duplication. 

(Revised manuscript, lines 30-31)

3. Thank you for including your ethics statement: All human myometrial tissues were obtained in accordance with the ethics committee of the Graduate School of Medicine, Kyoto University after written consent (G1149).

>>> The full name of the ethics committee of our facility is “Kyoto University Graduate School and Faculty of Medicine, Kyoto University Hospital Ethics Committee.” This information was added to the revised Materials and Methods section.

(Revised manuscript, lines 102-103)

>>> Original uncropped and unadjusted images of immunoblots were added to the Supporting information files of the revised manuscript. 

>>> Thank you for noting this. We included all supplementary data in the Supporting information files of the revised manuscript. An immunofluorescent image of PAR1 expression in a non-pregnant uterus was also added to revised Fig. 1B.

>>> We included captions for the Supporting information files at the end of our revised manuscript. The Supplementary movie is available in the public repository Zenodo. The method was moved to the revised Materials and Methods section. 

Reviewer #1: IS THE MANUSCRIPT TECHNICALLY SOUND, AND DO THE DATA SUPPORT THE CONCLUSIONS?

The manuscript follows a logical approach of using the collagen lattice assay to measure contractility induced in primary myometrial cells following treatment with thrombin. The extent of contraction is quantitated by measuring the area of lattice using Image J software.

Figure 1 demonstrates that both thrombin and its receptor, PAR1, are present in pregnant and non-pregnant human myometrium.

Figure 2 demonstrates that thrombin induces contraction of primary myometrial cells, and that activation of PAR1, by means of the activating peptide, also induces contraction. Furthermore, the PAR1 inhibitor, SCH79797, inhibited the vast majority (but not all) of the thrombin-induced contraction.

7. Curiously, page 9 line 7 indicates that the primary myometrial cells utilised to generate the data in Figure 2 were derived from non-pregnant myometrium, rather than pregnant myometrium. Is this correct or should this say.

>>> Thank you for these valuable comments. In our facility, cases of hysterectomy during pregnancy are rare, and sampling of myometrial strips at cesarean incisions is not performed for fear of increasing bleeding. Therefore, we do not have pregnant myometrial tissues readily available. Luckily, we had a rare case of hysterectomy due to cervical cancer Ib1 in pregnancy during this revision period (34 weeks of gestation), with permission of the Ethics Committee of Kyoto University Hospital. We performed a contraction assay and similar contraction by thrombin was reproduced as in non-pregnant myometrium (Fig. S2A). PTGS2 mRNA was also increased by thrombin in pregnant myometrium (Fig. S2B). These findings suggest that the use of non-pregnant myometrial cells is justified for studying myometrial contraction. We are planning to use pregnant myometrium in future experiments.

Figure 3 demonstrates that thrombin induces MLC-ser19 phosphorylation, and that this phosphorylation is inhibited by the PAR1 inhibitor. Densitometric

Figure 4 provides evidence that thrombin-induced contraction of primary myometrial cells is mediated by myosin light chain kinase and ROCK, as evidenced by the MLCK inhibitor (ML-7) and the ROCK inhibitor (Y-27632) both partially inhibiting thrombin-induced MLC-ser19 phosphorylation. Additionally, Panel D demonstrates that thrombin does not promote myometrial contraction by inhibiting (phosphorylating) the regulatory subunit of myosin phosphatase, MYPT1.

Figure 5 demonstrates that thrombin-induced contraction of primary myometrial cells is associated with (i) significantly upregulated expression of PTGS2 (COX2) after 4 h, (ii) significantly increased production of PGF2 alpha after 4 h, with maximum levels observed after 24 h, (iii) significantly upregulated expression of IL1B, (iv) significantly down-regulated expression of both PTGER3 (EP3) and PTGFR (FP), and (v) no change in expression of GJA1 (CX43), OXTR or PTGER1 (EP1). Additionally, panels E and F demonstrate that indomethacin partially inhibits the thrombin-induced contraction, demonstrating that thrombin-induced contraction is in-part indirect via stimulating prostaglandin production.

Figure 6 demonstrates that thrombin-induced contraction of primary myometrial cells is partially inhibited by pre-treating the cells with progesterone (P4) or medroxyprogesterone acetate (MPA).

The conclusions drawn are appropriate for the data, however, questions remain as to the whether the data is comprised of technical replicates or biological replicates.

HAS THE STATISTICAL ANALYSIS BEEN PERFORMED APPROPRIATELY AND RIGOROUSLY?

8. Further explanation is required as to what is being considered a replicate. For instance, the Figure 2 legend says “n=3 in each group. Experiment was repeated with cells from another preparation with identical results”. This reads as though the study was performed with 3 technical replicates (ie myocytes from 1 woman plated across multiple wells and then the treatment applied to 3 of those wells), and then repeated in only one other woman. If so, this would mean that only n=2 biological replicates have been examined (with 3 technical replicates for each biological replicate). The minimum requirement is n=3 biological replicates, so if n=3 biological replicates have indeed been performed, this needs to be explained clearly in the materials and methods.

>>> Thank you for your valuable comments. All of the experiments were repeated at least three times. Each experiment of the collagen lattice assay, qPCR, and ELISA was performed in triplicate. These replicated data are shown in the Figures of Replicated Experiments in the Supporting information files.

HAVE THE AUTHORS MADE ALL DATA UNDERLYING THE FINDINGS IN THEIR MANUSCRIPT FULLY AVAILABLE?

9. Uncropped versions of the western blots have been made available in the supplementary data, however, only it appears that data may only be available for n=1 biological replicate. Additionally, all the biological replicates for the collagen lattice assays (minimum required is n=3) do not appear to be available.

>>> As mentioned above, immunoblots were repeated three times. Representative blots are shown in figures, and the other replicated data are available in the Figures of Replicated Experiments. Collagen lattice assay was also repeated at least three times. 

IS THE MANUSCRIPT PRESENTED IN AN INTELLIGIBLE FASHION AND WRITTEN IN STANDARD ENGLISH?

10. English quality is quite good, however, there are some minor grammatical errors that should be addressed. Some of these issues are outlined below, but additional proof-reading is required with attention to the correct placement of commas.

>>> Thank you for your useful comment. The revised manuscript was sent to a professional native English-speaking scientific proofreader. 

Requested changes and corrections:

Introduction:

11. There is no detail provided in relation to the Image J measurement. Although this is a standard.

>>> The gel area was calculated by Image J software. The outside of a gel was manually traced by the “Polygon selection” tool and the area was calculated by the “Measure” tool. The unit of analyzed area was reflected as pixels. An example of the calculation procedure is shown below. This explanation was added to the Materials and Methods section of the revised manuscript.

(Revised manuscript, lines 122-125)

Materials and Methods:

12. Page 5, line 23: “and then added thrombin” should be “and then thrombin was added”

>>> The sentence was revised as per your suggestion. 

(Revised manuscript, line 118)

13. Correct SI unit for litre is ‘L’, not ‘l’. Apply throughout entire manuscript, including figures.

>>> Thank you for this useful comment. This correction was made throughout the revised manuscript. 

14. There is no detail provided in relation to the Image J measurement. Although this is a standard assay, additional information as to how the Image J analysis was performed would be welcomed to explain the result. For instance, in most graphs showing the results of the collagen assays, the mean relative pixel area is not 1.0, which indicates that the collagen discs did not occupy 100% of the area measured. Presumably the area measured was the size of the well? Such information should be conveyed as it affects the interpretation of the results (i.e. the extent of thrombin-induced contraction is not a percentage of the pixel area of the untreated control).

>>> Thank you for this useful comment. As mentioned above, the outside of each gel was manually traced by the “Polygon selection” tool of Image J and the area was analyzed by the “Measure” tool. The unit of area was reflected as pixels so the area of control is not necessarily 1.0. The area was not compared to the well size. 

15. It should be possible to fully understand the experiments conducted and the results obtained without having to refer to the figures. As such, in the materials and methods, please provide the concentrations of treatments in-text (for example, page 5, line 21).

>>> Thank you for this valuable comment. Detailed methods such as the concentration of reagents and treatment time are described for collagen lattice assay, immunoblots, and qPCR in the Material and Methods section.

16. Page 5, line 21: provide O2 and CO2 percentages.

>>> The cell culture incubation condition of oxygen (20%) and CO2 (5%) was added to the revised manuscript. 

(Revised manuscript, lines 93-94)

17. There is no mention of protein quantitation method and no details provided for SDS-PAGE, such as the amount of protein loaded per lane (supplementary figures indicate that 20 ug per lane was loaded, however, this information should be provided in-text in the methods). Type of membrane used for western blotting not indicated (PVDF or nitrocellulose?).

>>> The detailed procedure of immunoblotting was described in the “Immunoblots” section. BCA assay was utilized to protein quantification. Twenty micrograms of protein were loaded in each lane. A PVDF membrane was used because the molecular weight of MLC was comparably small (approximately 20 kDa). 

(Revised manuscript, lines 144-148)

18. Antibody dilutions are provided, however, this does not provide any indication of the quantity of antibody used, as different antibodies come at different dilutions. While it is common practice to merely report antibody dilutions, the authors are encouraged to additionally report the actual final antibody concentration utilised (i.e. 0.1 ug/mL in 20 mL of TBST). Not essential but encouraged.

>>> The exact antibody concentration was not available in the datasheet from Cell Signaling and Abcam so we described the dilution of antibody only (1:1000). All of the first antibodies were diluted into 5 mL of 5% BSA/TBST. The second antibody was diluted to 10 mL of 5% BSA/TBST. This information was added to the revised Immunoblots section.

(Revised manuscript, lines 150-159)

Results:

19. In the results, the collagen lattice assays graphs convey the reduction in relative pixel area, however, nowhere in the manuscript are the actual quantitated values provided. For each of the collagen lattice assay graphs, please report the mean pixel area (with +/- SD) for the different treatments in-text with the results. Additionally, the figure legends indicate that “*, p < 0.05, and **, p <0.01”, however, please provide the p-values in-text during the results. For example: Thrombin treatment significantly decreased relative pixel area compared to control (0.95 +/- 0.02 versus 0.6 +/- 0.03; p=0.03).

>>> We appreciate this useful comment. The mean pixels and SD in collagen lattice assay were described in the Results section. A p value was also added to the manuscript.

20. In the legend for Figure 5, Section C states “2U/ml of thrombin for 12 h”, however, the X-axis of panel C (which displays the effect of thrombin on PGF2 alpha levels) appears to show thrombin treatments at 1 and 2 U/mL.

>>> Thank you for pointing out this error. The legend for Figure 5 was corrected in the revised manuscript. New data on PGE2 concentration by thrombin treatment was also added.

21. Page 8, line 23-24: “Although not observed in the center of hemorrhage, PAR1 was expressed in myometrial cells peripheral to the bleeding site”. FYI: This is an interesting observation that leaves me curious as to why that may be.

>>> Microscopically, the structure of the myometrial tissue in the center of the bleeding site was devastated, probably due to protease activity and pressures resulting from hemorrhage. In contrast, the myometrial tissue peripheral to the bleeding site was intact. This is why PAR1 was expressed at the intact myometrium, peripheral to the bleeding site.

(Revised manuscript, lines 196-197)

22. Page11, line 10: ‘media of thrombin-treated…’ instead of ‘media by thrombin-treated…’

>>> The sentence was changed as per your useful suggestion. Thank you.

23. Number of replicates are not adequately identified throughout the results, either in -text or in the figure legends. For example, the legends for Figures 3 and 4 do not indicate how many biological replicates were performed for the western blots (i.e. was only 1 western blot performed for 1 woman/biological replicate, or was the blotting performed for primary myocytes isolated from n=3 different women?) or immunocytochemistry.

>>> All of the experiments were repeated three times and the data are shown in the Figures of Replicated Experiments. In the replicates, the primary myometrial cells were derived from the uteruses of three different women. The figure legends were thoroughly revised.

24. Page21, Figure 2 legend: Please replace “identical results” with “consistent results”, as it is impossible that identical results were observed across replicates.

>>> The Figure legends were thoroughly revised. We removed “identical results.” Consistent results were obtained and shown in the Figures of Replicated Experiments of the revised manuscript. 

General:

25. The throughout the entire manuscript and figures, authors are encouraged to use the correct protein names; i.e. Prostaglandin-Endoperoxide Synthase 2 (gene name: ‘PTGS2’), rather than cyclooxygenase (‘COX2’), with gene names in all-caps and non-italicised.

>>> Thank you for your suggestion. The Gene name COX2 was changed to PTGS2 throughout the revised manuscript and figures.

26. The throughout the entire manuscript and figures, the authors are encouraged to use the correct gene names (presented in app-caps and italicised). COX2 = PTGS2, CX43 = GJA1, EP1 = PTGER1, EP3 = PTGER3, FP = PTGFR. Refer to genecards.org.

>>> We appreciate this useful suggestion. All of the gene names were changed to the correct names. 

27. Grammar: Use ‘protease-activated receptor 1’, not ‘protease-activated receptor-1’.

>>> “Protease activated receptor-1” was changed to “protease-activated receptor 1” throughout the revised manuscript. Thank you.

28. ‘PAR-1’ and ‘PAR1’ used interchangeably. Additionally, ‘protease-activated receptor 1 is firdst used in the abstract and the abbreviation (PAR1) is provided. In-text, however, the PAR1 abbreviation is used without providing the full name (page 5, line 2).

>>> We removed the hyphen from “PAR-1”, and all the words were corrected to “PAR1” throughout the revised manuscript.

29. Page 14, line 12: what is meant by “activity”? This is vague and needs clarification.

>>> Thank you for this useful comment. The protease activity of thrombin was increased by an increase in the amnion of preterm labor (Figure below, Mogami et al. JBC 2014). We changed this to “protease activity of thrombin” in the revised manuscript. 

(Revised manuscript, line 429)

30. Page 14, line 13-15: “Therefore, inhibition of thrombin signaling pathways has potential to treat preterm labor associated “without” intrauterine bleeding.”. Please correct this sentence: “Therefore, inhibition of thrombin signaling pathways has potential to treat labor that is not associated with intrauterine bleeding”.

>>> Thank you for this useful comment. We revised the last sentence as per your suggestion.

(Revised manuscript, lines 431-432)

Comments:

31. The primary myometrial cells utilised were isolated from myometrium obtained from non-pregnant premenopausal women. The manuscript would benefit from authors commencing the Discussion by justifying their case for not using pregnant myometrium and provide a brief argument as to why findings gleaned from nonpregnant primary uterine myocytes still provide valid insight into the mechanisms underpinning premature birth.

>>> Thank you for this valuable comment. Ideally, the experiments should be performed using pregnant uterus. As added in the revised Materials and Methods section, however, hysterectomies in pregnant women are very rare in our facility, and we do not obtain myometrial strips during cesarean births. Fortunately, we were able to obtain myometrial tissue from a rare case of cesarean hysterectomy due to a pregnancy complicated by early stage Ib1 cervical cancer. We isolated myometrial cells from the pregnant uterine corpus and observed that contraction of the myometrial cells was similar to those from non-pregnant uterus (Fig. S2). In addition, thrombin receptor PAR1 was expressed in both pregnant and non-pregnant myometrium (Fig. 1). O'Sullivan et al. showed that thrombin and PAR1 activating peptide exerted a stimulatory effect on uterine contraction in both pregnant and non-pregnant myometrial tissues (O'Sullivan CJ, 2004). They also reported that there was no significant difference in sensitivity to thrombin between pregnant and non-pregnant myometrium. Therefore, as the reviewer suggested, we believe that our data obtained from non-pregnant myometrium still provide valid insight into the mechanisms of preterm birth induced by intrauterine bleeding.

(The data on pregnant myometrium was added to the revised manuscript and Fig. S2.)

(Revised manuscript, lines 83-86 and lines 342-352)

32. There is no mention of existing literature that examines the role/regulation of thrombin in normal labour. Ie. What happens to thrombin levels during normal labour? These studies should be briefly mentioned as existing evidence indicates that thrombin activity rises during normal, uncomplicated labour, suggesting that thrombin activity is a normal component labour. Additionally, this has implications for the author’s claim that (page 3, line 20) “Collectively, these data suggest that thrombin is a pathogenic factor in initiation of preterm birth”, which suggests a causative role in preterm birth. It is possible that thrombin activity in indeed causative of premature labour, as supported by thrombin by inducing PTB in mice, but it is also possible that thrombin activity rises as a result of the woman being in labour, whether the labour is at term or preterm. As such, the manuscript would benefit by highlighting that the key underlying issue is that dysregulation of thrombin activity has the potential to cause the premature onset of labour, leading to preterm birth.

>>> Thank you for these valuable comments. As the reviewer suggested, thrombin-antithrombin complex (TAT) levels increased during normal pregnancy [1]. The mean TAT level is 1.9 ± 0.3 μg/L in non-pregnant women regardless of menstrual cycle phase, but increases to 16.0 ± 2.8 μg/L in the 2nd trimester, 21.5 ± 11.9 μg/L at term, and 28.6 ± 12.8 μg/L in the 2nd stage of labor [2]. In addition, TAT further increases in the 3rd stage of labor upon separation of the placenta [2, 3]. This is due to the release of thromboplastic substances at the site of placental separation. It seems that released thrombin from the placenta-attached site to the uterus facilitates the contraction of myometrium (uterine involution), and prevents massive hemorrhage at parturition. Therefore, the abnormal increase of thrombin mid-trimester is a pathological process that leads to preterm birth whereas the increase of maternal circulating thrombin at the 3rd stage of labor and parturition is a physiological process. These physiological roles of thrombin during normal labor were included in the manuscript. 

(Revised manuscript, lines 39-42). 

33. There is no discussion on how expression of the genes encoding thrombin (F2) and PAR1 (F2R) are regulated in the myometrium. The manuscript would benefit from discussing what is known about F2 (primarily) and F2R (secondly) expression in the myometrium (and other tissues), and how dysregulation of expression of these genes could be causal to preterm birth.

>>> Thrombin is almost exclusively synthesized in the liver (https://www.ncbi.nlm.nih.gov/gene/2147), and we confirmed that pregnant myometrium did not express thrombin (F2) in immunofluorescence (Fig. S1). An explanation of PAR1 was added to the Discussion section of the revised manuscript. 

O’Brien et al. showed that PAR1 (F2R) was expressed in both pregnant and non-pregnant myometrium, which is compatible with our data from immunofluorescence (Fig. 1 and 3); they also showed that expression is higher during pregnancy compared to non-pregnancy [4]. Further, they demonstrated that PAR1 expression increased 9-fold during labor in human myometrium compared to the state of not in labor, indicating that the sensitivity to thrombin is increased during labor. In addition to the physiological increase in thrombin, increased expression of PAR1 would contribute to stronger contractions during labor and puerperium. This description was added to the revised Discussion section.

(Revised manuscript, lines 416-426). 

34. The authors suggest a potential clinical role for progesterone in inhibiting preterm labour complicated by intrauterine bleeding, based on the ability of progesterone to partially inhibit thrombin-induced myometrial contraction. Existing literature indicates that the relaxatory effects of progesterone are mediated by PR-B, but also that prostaglandins increase the PR-A/PR-B ratio. Given that the authors present data indicating that thrombin induces production of prostaglandins, the manuscript would be benefit from discussing how thombin-induced prostaglandin production may affect the PR-A/PR-B ratio, and how this may affect the clinical usefulness of progesterone therapy.

>>> Thank you for this useful comment. As you mentioned, progesterone receptors are key molecules to regulate the contraction of myometrium during pregnancy. The PR-A/PR-B ratio increased in myometrium during labor, which inhibits the action of progesterone and leads to successful delivery [5, 6]. We added experiments examining how thrombin and prostaglandins regulate the mRNA expressions of progesterone receptors (PgR-A and PgR-B). Thrombin tended to decrease PgR-A and PgR-B mRNA levels so that the total abundance of PgRs was decreased. Consequently, the PgR-A/PgR-B ratio was not changed by thrombin. Previously, Madsen G et al. showed that PGE2 and PGF2α increased PR-B expression and the PR-A/PR-B ratio was increased by comparably lower doses of prostaglandins [7]. We tested this, but neither PGE2 nor PGF2α regulated PgR-A and PgR-B mRNA. The reason for this difference between Madsen et al.’s results and ours is unclear, but Madsen et al. also showed that the PgR-A/PgR-B mRNA ratio tended to return to the basal level when prostaglandin concentrations increased. We used a comparably high dose of prostaglandins (10 nM) in our experiments, so this might be a reason why the PgR-A to PgR-B mRNA ratio was not altered in our study. Although the PgR-A/PgR-B ratio was not regulated, thrombin strengthens myometrial contractions by reducing the progesterone responsiveness caused by downregulation of progesterone receptor expression. Further study is necessary for clarifying the precise regulation of PR by thrombin.

(Revised manuscript, lines 320-333, and 395-408)

35. I’m not sure whether it is an artefact of the production of the draft version of the manuscript, but the figures appear to be low resolution. There is extensive pixilation. Hopefully the final version will have figures of greater clarity.

>>> The original figures are high-resolution. The publisher instructed us to submit figures as PDF, and the converted PDFs on the publisher’s web site appear low resolution. We are confident that the final version will be clearer. 

Overall, this study presents some interesting findings that, if true, will make a valuable contribution to advancing our understanding of the mechanisms contributing to the syndrome of preterm birth. However, at this stage, there appears to be insufficient biological replicates, with the results instead based on the statistical analysis of technical replicates (if this reviewer’s interpretation of the data is incorrect, then could the authors please provide some clarity). If the findings persist following the incorporation of additional biological replicates, the study will make a valuable contribution to the field.

Reviewer #2: This interesting paper explores the ability of thrombin, released most often during intrauterine hemorrhage, on its ability to stimulate other mediators (IL-1beta, prostaglandins, COX-2) via the PAR-1 receptor and MLCK intracellular pathways and the ability of progesterone to block these.

The writing of the manuscript, techniques, data analysis and figures are very good. I found just one typo in the MS (‘as’ should be replaced with ‘at’ in one figure legend).

My major recommendation though is that the there are uncompleted experiments that will improve the information shared and conclusions reached. These begin by noting that thrombin increased COX-2 expression and PGF2alpha output, but that expression of the FP receptor decreased. The authors did not address this important observation. I am quite certain the increased PGF2a produced cause the down regulation of its receptor, FP. Although PGE2 was not assayed, I’m also certain that thrombin increased it, too, and its higher concentrations led to the down regulation of the EP3 receptor noted in the figures. Again, this was not addressed very well by the authors.

35. I am suggesting that the authors perform a few more simple experiments to address these interesting observations in order to explain their results better. One of the questions that derives, is does thrombin stimulate a decrease in the expression of EP3 and FP directly or is this due to increased prostaglandins that down regulate their receptors. Thus if indomethacin or a COX-2 specific inhibitor is co-administered with thrombin, what is the outcome with the PG receptors? I would suggest that in this case, there is no down-regulation of EP3 or FP. If true, then administration of PGE2 or PGF2a and assessing expression of EP3 and FP should confirm that PGs down regulate their receptor expression. I propose that thrombin increases IL-1b and that IL-1b increases expression of EP3 and FP as well as COX-2. The reason that thrombin stimulates contractions is that the action of PGE2 and PGF2a is not lost (due to large down regulation of their receptors) because IL1b is stimulating their up-regulation. Hence they are present enough to respond to their agonists and effect contraction.

>>> Thank you very much for these valuable suggestions. We assayed the PGE2 concentration in the conditioned media of thrombin-treated myometrial cells. As you expected, thrombin increased PGE2 dose- and time-dependently (Fig. 5E and 5F). This is a very important finding. 

In order to investigate the mechanism of how the expression of EP3 and FP was decreased and PTGS2 increased, we performed additional experiments: 1) IL-1β treatment of myometrial cells, 2) Thrombin with pretreatment of indomethacin, and 3) PGE2 and PGF2α treatment. From these experiments, we concluded that:

1) IL-1β increased PTGS2 mRNA, so the increase of PGE2 and PGF2α was mediated by thrombin-induced IL-1β (Fig. 6B). In addition, PGE2, but not PGF2α, increased IL1B mRNA (Fig. 6D). This suggests that PGE2 is increased by positive feedback through IL-1β (Fig. 6F). 

2) Thrombin and IL-1β decreased PTGER3 mRNA, and pretreatment of indomethacin recovered the decrease in PTGER3 mRNA by thrombin and IL-1β (Fig. 6C). In addition, both PGE2 and PGF2α decreased PTGER3 mRNA (Fig. 6D and 6E). These findings suggest that the thrombin-induced decrease of PTGER3 mRNA was mediated by PGE2 and PGF2α via IL-1β (Fig. 6F). 

3) In contrast, IL-1β did not decrease PTGFR mRNA (Fig. 6B), and indomethacin did not recover the decrease in PTGFR mRNA by thrombin (Fig. 6C). In addition, neither PGE2 nor PGF2α changed PTGFR mRNA (Fig. 6D and 6E). These findings suggest that the thrombin-induced decrease of PTGFR mRNA was not regulated by PGE2 or PGF2α, but by another unknown pathway (Fig. 6F). 

 We summarized the relationship of thrombin, IL-1β, prostaglandins, and the regulation of prostaglandin receptors in Fig. 6F. The data were added to the Results section of the revised manuscript. 

(Revised manuscript, lines 294-304). 

36. However, it is noted from the data that thrombin administration also increases IL-1b. But IL-1b on its own can stimulate COX-2 expression and it increases the expression of FP (see Mol Hum Reprod. 2015 Jul;21(7):603-14.). These should be confirmed in these cells. If true then the question derives how can thrombin stimulate myometrial contraction if it leads to the down regulation of FP or EP3 as shown by the limited data presented? In this case, stimulate cells with IL-1b in the absence or presence of indomethacin or a COX-2 inhibitor. I suspect there will be a large increase in EP3 and FP expression when COX-2 is inhibited, and less or even a decrease in expression when COX-2 activity is intact.

>>> Thank you for your useful comments. As you suggested, IL-1β robustly increased PTGS2 mRNA (Fig 6B), and IL-1β decreased PTGER3 mRNA but it did not regulate PTGFR mRNA expression (Fig. 6B and figures below). 

 Pretreatment of indomethacin partially alleviated the decrease in PTGER3 mRNA by IL-1β (Fig. 6C). Therefore, we propose that a decrease in EP3 receptor by thrombin is mediated by prostaglandins through IL-1β, whereas regulation of FP receptor is by another pathway. The physiological meaning for why thrombin can keep stimulating contraction of myometrium while suppressing the expressions of prostaglandin receptors is unclear. The elucidation of the precise mechanism of how thrombin downregulates prostaglandin receptors is our next area of study. 

37. Finally, one begins to wonder where the action of progesterone or MPA is at. Is it directed at inhibiting COX-2 action directly, or is it directed at inhibiting expression of IL-1beta or even directed at PAR-1 expression? I strongly suggest to the investigators and to the Editor to include these important experiments in this manuscript to explain more completely the relationships hinted at by the data that is presented.

>>> Thank you very much for this valuable comment. We added an experiment involving thrombin with pretreatment of 1 μM progesterone. The increase in IL1B mRNA was suppressed by pretreatment of 1 μM of progesterone (Fig. 7B). Thrombin receptor PAR1 (F2R) was not regulated by thrombin, but progesterone repressed the F2R mRNA expression (Fig. 7B). Therefore, as Reviewer 2 commented, the function of progesterone was mediated by 1) suppression of PTGS2 mRNA expression, 2) suppression of IL-1β, and 3) decrease in F2R mRNA expression, even without thrombin treatment. All of these changes by progesterone contribute to the relaxation of myometrium. Data and discussion were added to the revised manuscript. 

(Revised manuscript, lines 316-318) 

References

1. Elovitz MA, Baron J, Phillippe M. The role of thrombin in preterm parturition. American journal of obstetrics and gynecology. 2001;185(5):1059-63. doi: 10.1067/mob.2001.117638. PubMed PMID: 11717633.

2. Uszynski M. Generation of thrombin in blood plasma of non-pregnant and pregnant women studied through concentration of thrombin-antithrombin III complexes. European journal of obstetrics, gynecology, and reproductive biology. 1997;75(2):127-31. doi: 10.1016/s0301-2115(97)00101-2. PubMed PMID: 9447363.

3. Yoshimura T, Ito M, Nakamura T, Okamura H. The influence of labor on thrombotic and fibrinolytic systems. European journal of obstetrics, gynecology, and reproductive biology. 1992;44(3):195-9. doi: 10.1016/0028-2243(92)90098-j. PubMed PMID: 1535054.

4. O'Brien M, Morrison JJ, Smith TJ. Expression of prothrombin and protease activated receptors in human myometrium during pregnancy and labor. Biology of reproduction. 2008;78(1):20-6. doi: 10.1095/biolreprod.107.062182. PubMed PMID: 17901076.

5. Merlino AA, Welsh TN, Tan H, Yi LJ, Cannon V, Mercer BM, et al. Nuclear progesterone receptors in the human pregnancy myometrium: evidence that parturition involves functional progesterone withdrawal mediated by increased expression of progesterone receptor-A. The Journal of clinical endocrinology and metabolism. 2007;92(5):1927-33. doi: 10.1210/jc.2007-0077. PubMed PMID: 17341556.

6. Mesiano S, Chan EC, Fitter JT, Kwek K, Yeo G, Smith R. Progesterone withdrawal and estrogen activation in human parturition are coordinated by progesterone receptor A expression in the myometrium. The Journal of clinical endocrinology and metabolism. 2002;87(6):2924-30. doi: 10.1210/jcem.87.6.8609. PubMed PMID: 12050275.

7. Madsen G, Zakar T, Ku CY, Sanborn BM, Smith R, Mesiano S. Prostaglandins differentially modulate progesterone receptor-A and -B expression in human myometrial cells: evidence for prostaglandin-induced functional progesterone withdrawal. The Journal of clinical endocrinology and metabolism. 2004;89(2):1010-3. doi: 10.1210/jc.2003-031037. PubMed PMID: 14764828.

---

## [Decision Letter · Decision Letter 1]

6 Apr 2020

Mechanisms of thrombin-Induced myometrial contractions: Potential targets of progesterone

PONE-D-19-21553R1

Dear Dr. Mogami,

We are pleased to inform you that your manuscript has been judged scientifically suitable for publication and will be formally accepted for publication once it complies with all outstanding technical requirements.

With kind regards,

Tamas Zakar

Academic Editor

PLOS ONE

Additional Editor Comments (optional):

Reviewers' comments:

Reviewer's Responses to Questions

**Comments to the Author**

1. If the authors have adequately addressed your comments raised in a previous round of review and you feel that this manuscript is now acceptable for publication, you may indicate that here to bypass the “Comments to the Author” section, enter your conflict of interest statement in the “Confidential to Editor” section, and submit your "Accept" recommendation.

Reviewer #1: All comments have been addressed

Reviewer #2: All comments have been addressed

2. Is the manuscript technically sound, and do the data support the conclusions?

Reviewer #1: Yes

Reviewer #2: Yes

3. Has the statistical analysis been performed appropriately and rigorously? 

Reviewer #1: Yes

Reviewer #2: Yes

4. Have the authors made all data underlying the findings in their manuscript fully available?

Reviewer #1: Yes

Reviewer #2: Yes

5. Is the manuscript presented in an intelligible fashion and written in standard English?

Reviewer #1: Yes

Reviewer #2: Yes

6. Review Comments to the Author

Reviewer #1: Thank you to the authors, who have answered or addressed all of the questions and feedback I raised. The edits have significantly clarified the manuscript and the additional data have increased its significance. The article will make a valuable contribution to the field. There are still some minor typographical inconsistencies (see below), however, I expect that these will be corrected during publication and proofing.

Minor issues:

• Line 417: '1.9' appears twice.

• Some inconsistency with the way the PG names are written, i.e. PGF2αlpha (symbol) vs PGF2α vs PGF2 α (symbol), and there’s also one instance of F2a instead of F2α (symbol). There’s also some inconsistency with PGE2.

Reviewer #2: Thank you for addressing our questions and suggestions so thoroughly. I especially appreciate the performance of extra experiments as suggested.

7. PLOS authors have the option to publish the peer review history of their article (what does this mean?). If published, this will include your full peer review and any attached files.

Reviewer #1: No

Reviewer #2: Yes: David M Olson

---

## [Editor Report · Acceptance letter]

17 Apr 2020

PONE-D-19-21553R1 

Mechanisms of thrombin-Induced myometrial contractions: Potential targets of progesterone 

Dear Dr. Mogami:

I am pleased to inform you that your manuscript has been deemed suitable for publication in PLOS ONE. Congratulations! Your manuscript is now with our production department. 

With kind regards,

on behalf of

Dr Tamas Zakar 

Academic Editor

PLOS ONE